# Barrel cortex plasticity after photothrombotic stroke involves potentiating responses of pre-existing circuits but not functional remapping to new circuits

William A. Zeiger[1✉], Máté Marosi[1,3], Satvir Saggi [1], Natalie Noble[1], Isa Samad[1] & Carlos Portera-Cailliau [1,2✉]

Recovery after stroke is thought to be mediated by adaptive circuit plasticity, whereby surviving neurons assume the roles of those that died. However, definitive longitudinal evidence of neurons changing their response selectivity after stroke is lacking. We sought to directly test whether such functional "remapping" occurs within mouse primary somatosensory cortex after a stroke that destroys the C1 barrel. Using in vivo calcium imaging to longitudinally record sensory-evoked activity under light anesthesia, we did not find any increase in the number of C1 whisker-responsive neurons in the adjacent, spared D3 barrel after stroke. To promote plasticity after stroke, we also plucked all whiskers except C1 (forced use therapy). This led to an increase in the reliability of sensory-evoked responses in C1 whisker-responsive neurons but did not increase the number of C1 whisker-responsive neurons in spared surround barrels over baseline levels. Our results argue against remapping of functionality after barrel cortex stroke, but support a circuit-based mechanism for how rehabilitation may improve recovery.

---

[1] Department of Neurology, David Geffen School of Medicine, University of California Los Angeles, Los Angeles, CA, USA. [2] Department of Neurobiology, David Geffen School of Medicine, University of California Los Angeles, Los Angeles, CA, USA. [3] Present address: Department of Pharmacology and Toxicology, University of Texas Medical Branch, Galveston, TX, USA. ✉email: wzeiger@mednet.ucla.edu; CPCailliau@mednet.ucla.edu

Stroke is the fifth leading cause of death and the leading cause of adult-onset disability in the US[1]. Many stroke patients exhibit partial spontaneous recovery of function, which can be improved with rehabilitation[2,3]. This suggests that the brain has endogenous mechanisms to restore lost functions. The prevailing hypothesis is that post-stroke plasticity involves a process of circuit remapping, in which neurons that survive the injury are recruited to take over the function of neurons that died[4,5]. However, even though functional remapping occurs in the healthy brain during learning[6] and in response to changes in sensory experience[7,8], irrefutable evidence for remapping of lost functionalities to new circuits in the context of stroke remains lacking.

Human studies with brain imaging or neurophysiology have revealed variable (and sometimes opposite) changes in brain activity or metabolism after stroke. Some of these changes, such as the apparent reorganization of cortical sensorimotor maps or differences in resting state functional connectivity (reviewed elsewhere[9]), have been interpreted as evidence of remapping. But they could also reflect normal variability across individuals, especially when considering that these imaging modalities could be influenced by altered hemodynamics post-stroke and that pre-stroke baseline data were not available. In mice, macroscopic brain imaging studies have similarly shown a range of alterations in cortical activity maps after stroke[10–13], but whether those changes were associated with functional recovery has not been established. In a landmark in vivo calcium imaging study often cited as the best evidence for remapping after stroke, some neurons in peri-infarct cortex were more broadly tuned after stroke compared to neurons in control mice without a stroke[14]. However, the fraction of neurons that had assumed new roles was rather small and, just like with the human studies, individual mice were not imaged longitudinally before and after stroke.

Further complicating our understanding of post-stroke plasticity, other studies have identified circuit alterations that would be expected to limit, rather than promote, remapping. In human stroke, compensation by the unaffected limb appears to induce maladaptive plasticity[15,16], perhaps via increased interhemispheric inhibition[17]. Pathological increases in inhibition after stroke have also been identified in animal models of stroke, and reducing this inhibition can promote recovery[18,19]. In addition, experience-dependent plasticity in both somatosensory and visual cortex is impaired following strokes targeted to the somatosensory cortex[20–22], which implies that the post-stroke environment is nonpermissive for plasticity. In summary, definitive evidence for functional remapping after stroke remains lacking, and whether all aspects of post-stroke plasticity are beneficial remains controversial.

Addressing these knowledge gaps about stroke plasticity requires longitudinal in vivo recordings of the activity of individual neurons before and after stroke. Toward that end, we performed in vivo intrinsic signal imaging (ISI) and two-photon (2P) calcium imaging of sensory-evoked responses before and after a photothrombotic (PT) stroke that was targeted to a specific barrel (C1) in the barrel field of primary somatosensory cortex (S1BF). We found no clear evidence of remapping of lost functionalities to new circuits in peri-infarct cortex. However, plucking all whiskers except the one corresponding to the infarcted barrel (as a forced use rehabilitative strategy) significantly enhanced the reliability of responses in neuronal ensembles that were already responsive to that whisker at baseline. Our results argue against the classic remapping model of stroke recovery (where surviving neurons/circuits can assume a new role), at least in the barrel cortex, and are instead more consistent with a model where spontaneous or rehabilitation-induced recovery involves potentiation of pre-existing circuits.

## Results

**A single-barrel-targeted cortical lesion model for testing circuit remapping after stroke.** To study neuronal plasticity and remapping at the local circuit level, before and after focal cortical lesions, we focused on the barrel field of primary somatosensory cortex (S1BF) for several reasons: (1) somatosensory deficits are common after stroke in humans[23,24]; (2) as nocturnal animals, mice preferentially rely on their whiskers to explore their surroundings; (3) S1BF exhibits significant experience-dependent plasticity, even in adult animals[8,25,26]; and (4) the dynamics of neuronal circuits in rodent S1BF are well characterized[27–30]. The S1BF exhibits precise somatotopic organization with each barrel receiving inputs primarily from its corresponding peripheral whisker (labeled by row and arc position, e.g., C1, D3, etc.)[28]. Importantly, recent studies have revealed that individual neuronal responses within a given barrel are heterogenous, with a small but substantial population of neurons that are preferentially tuned not to the principal whisker of that barrel, but to adjacent whiskers[31]. Furthermore, this "salt and pepper" distribution of whisker selectivity is highly dynamic and can adapt to changes in sensory experience[32]. In light of the prevailing theory of stroke remapping[4], given the barrel cortex's inherent capacity for plasticity and the presence of surround whisker tuned cells, we expected to find that after strokes targeted to the C1 barrel, neurons in adjacent spared barrels would take over the lost functionality.

**No evidence for remapping of individual whisker function during spontaneous recovery.** To investigate remapping, we adapted the PT stroke model to target the C1 barrel in S1BF, eliminating the C1 cortical activity map (which includes the C1 barrel and portions of the immediately surrounding barrels), while sparing most other neighboring barrels of the S1BF, i.e., the peri-infarct region (Fig. 1 and Supplementary Fig. 1). The resulting infarcts (Fig. 1c) were meant to mimic strokes in humans, which tend to be very small, with an average infarct size of <5% of total brain volume, and frequently <1% (refs. [33–37]). For an initial set of experiments, we implanted cranial windows over S1BF in young adult mice and targeted PT strokes specifically to the C1 barrel (Fig. 1). We used ISI[38] to identify single whisker-evoked maps in S1BF at the pre-stroke baseline and longitudinally for over 1 month following stroke (Fig. 1b). We used 100 Hz whisker stimulation for ISI, because we found it produced more robust signals without any change in map location compared to 10 Hz stimulation (Supplementary Fig. 2A). Before stroke, we confirmed that ISI activity maps for the C1 and D3 barrels were spatially distinct, with the D3 map located anterior to the C1 map, as expected (Fig. 1d and Supplementary Fig. 2B–E). In sham control animals (exposed to laser light but not Rose Bengal; see "Methods"), the C1 and D3 whisker-evoked maps remained stable in size and location throughout the duration of imaging (Fig. 1d, upper panel). In stroke animals, we could discern the PT infarct within the C1 barrel as a pale area, where the ISI map was previously located (Fig. 1d, lower panel). Following stroke, D3 whisker-evoked activity maps remained unaffected (Fig. 1d, lower panel), whereas the C1 whisker-evoked activity map disappeared and did not recover even 1 month after stroke (Fig. 1e, f and Supplementary Fig. 3).

There are several reasons why the C1 whisker-evoked ISI activity map might not re-emerge after stroke. One possibility is simply that remapping does not occur after stroke (i.e., spared neurons in peri-infarct cortex cannot assume the lost functions). Another reason is that ISI may lack the resolution to detect the activity of small numbers of neurons (scattered over a large cortical area or multiple layers) that exhibit compensatory

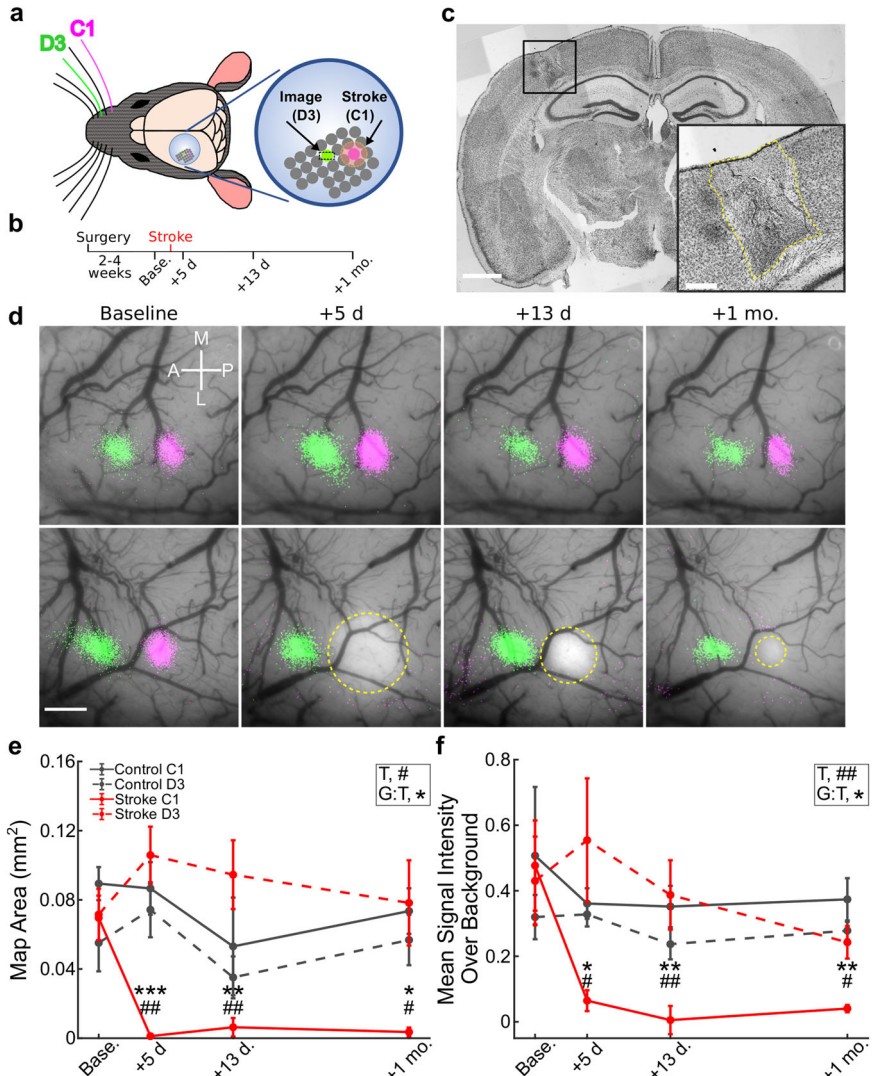

**Fig. 1 Intrinsic signal imaging reveals no evidence of macroscopic remapping after C1-targeted photothrombotic strokes. a** Left: schematic of cranial window placement and whiskers for stimulation. Right: enlarged view showing locations of the stroke (C1, magenta) and spared barrel (D3, green) highlighted. **b** Experimental timeline: ISI and 2P imaging were carried out at baseline (before stroke) and +5 days, +13 days, and +1 month after stroke. **c** Cresyl violet-stained coronal section from a representative mouse 5 days post-stroke. Scale bar = 1 mm. Inset: higher magnification view of infarct core, outlined in yellow. Scale bar = 250 μm. **d** ISI maps elicited by stimulation of C1 (magenta) and D3 (green) whiskers overlaid on photographs of the cranial window in representative sham control (top) and stroke (bottom) mice before and after stroke. Pale areas (yellow outline) represent the infarct. Note that the infarct size appears smaller over time as a result of the disappearance of acute tissue edema and tissue involution/scarring. Scale bar = 0.5 mm. **e** Quantification of C1 (solid lines) and D3 (dashed lines) whisker-evoked ISI map area size throughout recovery in control (gray) vs. stroke (red) animals ($n = 4$ and 6, respectively). Two-way repeated measures ANOVA, main effects of timepoint (T, #$p = 0.042$) and group-by-timepoint interaction (G:T, *$p = 0.014$). Significance for multiple comparison testing using the Tukey-Kramer procedure of timepoints (compared to baseline; #$p < 0.05$; ##$p < 0.01$); and group-by-timepoint interactions (comparing C1 and D3 map area size in the stroke group; *$p < 0.05$; **$p < 0.01$; ***$p < 0.01$). Data in **e** and **f** represent mean ± s.e.m. **f** Quantification of C1 (solid lines) and D3 (dashed lines) whisker-evoked ISI mean signal intensity over background signal throughout recovery in control (gray) vs. stroke (red) animals ($n = 4$ and 6, respectively). Two-way repeated measures ANOVA, main effects of timepoint (T, ##$p = 0.003$) and group-by-timepoint interaction (G:T, *$p = 0.03$). Significance for multiple comparison testing using the Tukey-Kramer procedure of timepoints (compared to baseline; #$p < 0.05$; ##$p < 0.01$;) and group-by-timepoint interactions (comparing C1 and D3 signal intensity in the stroke group; *$p < 0.05$; **$p < 0.01$).

plasticity. Alternatively, changes in blood flow and oxygen content after stroke (i.e., impaired neurovascular coupling[39]) may affect our ability to detect new ISI maps after stroke. To address these concerns, we also used in vivo 2P calcium imaging to record the activity of L2/3 neurons in the C1 and D3 barrels (previously identified by ISI) of Thy1-GCaMP6s mice in response to 20 sequential stimulations of the C1 or D3 whiskers (see "Methods" and Fig. 2). Before stroke, 32.6 ± 4.3% of L2/3 neurons in the D3 barrel responded to stimulation of their principal D3

whisker with stimulus-locked responses (Fig. 2c), consistent with our previous studies[40]. In contrast, only 13.9 ± 4.1% of L2/3 neurons in the D3 barrel responded to the surrounding C1 whisker (Fig. 2d), consistent with other calcium imaging studies[31]. We predicted that the percentage of C1-responsive neurons within the D3 barrel would increase after stroke, which would reflect remapping of the lost representation of the C1 whisker. However, we found instead that the percentage of L2/3 neurons whose calcium transients were stimulus-locked to

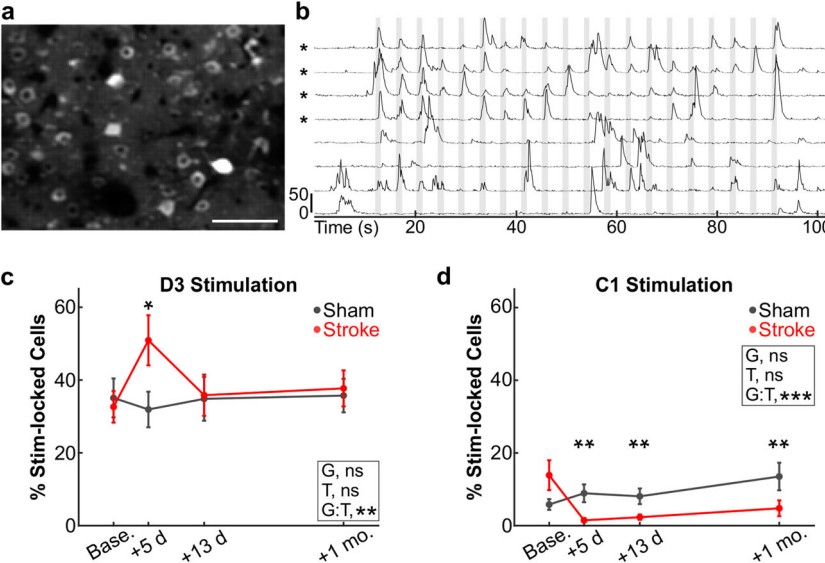

**Fig. 2 No increase in the percentage of C1 whisker-responsive neurons in peri-infarct cortex after C1-targeted stroke. a** Maximum intensity projection image (representative across all 15 animals imaged in this experiment) of GCaMP6s fluorescence in the D3 barrel of a Thy1-GCaMP6s mouse, before stroke. Scale bar = 100 μm. **b** Representative traces of GCaMP6s fluorescence signal intensity from four stimulus-locked (indicated with asterisks) and four non-stimulus-locked L2/3 neurons in response to whisker stimulation (10 Hz). Scale bar indicates Z-score change of 50. Vertical gray bars indicate epochs of whisker stimulation (1 s long, 3 s interstimulus interval). **c** Percentage of L2/3 neurons in the D3 barrel showing responses that were stimulus-locked to D3 whisker stimulation for mice that received a sham procedure (gray) or a photothrombotic stroke (red) to the C1 barrel. Individual timepoints are means ± s.e.m. GLME binomial model, ANOVA for fixed effects of group (G, $p = 0.736$), timepoint (T, $p = 0.803$), and group-by-timepoint interaction (G:T, **$p = 0.009$) are indicated on the respective plots. **d** Percentage of L2/3 neurons in the D3 barrel showing responses that were stimulus-locked to C1 whisker stimulation. GLME binomial model, ANOVA for fixed effects of group (G, $p = 0.154$), timepoint (T, $p = 0.236$), and group-by-timepoint interaction (G:T, ***$p < 0.001$) are indicated on the respective plots. Data in **c** and **d** represent mean ± s.e.m. $N = 6$ and 9 mice for sham (gray) and stroke (red), respectively, except $n = 8$ for stroke mice at +13 days in **d** and $n = 6$ for stroke mice at +1 month in **c** and **d**. Significance for individual coefficients for G:T, corrected using Benjamini and Hochberg's method, are indicated over corresponding data points (*$p < 0.05$; **$p < 0.01$).

stimulation of the C1 whisker (see "Methods") actually decreased significantly to <2% at 5 days post-stroke and remained below pre-stroke levels up to 1 month post-stroke (Fig. 2d). Interestingly, the percentage of D3 barrel neurons responding to the D3 whisker was transiently increased above baseline at 5 days post-stroke (perhaps due to loss of intracortical inhibitory inputs from the lesioned C1 barrel[41]), but returned to baseline thereafter (Fig. 2c). Sensory-evoked responses in D3 whisker-responsive neurons remained unchanged (Supplementary Fig. 4). In sham control animals, the percentage of C1 and D3 whisker-responsive cells remained stable across imaging sessions (Fig. 2c, d, black lines).

As a complementary approach to investigate the remapping hypothesis, we employed an activity-dependent labeling strategy using TRAP (targeted recombination in active populations) to identify new neurons that responded to the C1 whisker in surrounding barrels. This approach has been used to permanently label whisker-responsive neurons in vivo[42]. We randomized double transgenic TRAP mice (cFos-CreER^T2 × Ai9) to receive either sham or PT strokes of the C1 barrel, and allowed them to recover spontaneously for 2 months. Mice then underwent trimming of all whiskers except the contralesional C1 whisker. The next day, mice were injected with the CreER^T2 ligand 4-hydroxytamoxifen and then allowed to explore an enriched environment in the dark for 6 h (see "Methods" and Fig. 3a). In sham animals, TRAP resulted in robust tdTomato expression in neurons within the C1 barrel, primarily in L2/3 and L4, as well as sparser labeling in other layers and in surrounding barrels (Fig. 3b, left panels). We then compared the number of fluorescently labeled neurons in the barrels surrounding the C1 barrel (or the infarct) in sham and stroke groups. Compared to sham controls, we did not find an increase in the total number of

C1 whisker-responsive neurons in stroke mice, even when looking at individual cortical layers (Fig. 3c). Taken together, our results so far do not support the theory of remapping of neuronal function after focal cortical injury, because we could not find more neurons that responded to the C1 whisker in surrounding barrels after stroke than before stroke.

**Forced use therapy via whisker plucking does not promote remapping of individual whisker function after stroke.** It is possible that circuit remapping can only occur under certain circumstances, in which additional plasticity mechanisms are engaged, like in the setting of rehabilitation[9,43]. We explored this possibility by utilizing whisker plucking, a well-established paradigm for inducing neuronal plasticity in S1BF. Whisker plucking promotes circuit remapping in the intact S1BF through changes in dendritic spines[44,45] and in the excitation/inhibition ratio[25,46], resulting in retuning of L2/3 neurons away from deprived whiskers and toward spared whiskers[32], and in expanded whisker-evoked ISI activity maps[38,47]. We also favored this approach because forced use of an affected limb is a common rehabilitative technique used in human stroke patients[2,48]. We reasoned that forced use of the C1 whisker after stroke should engage inherent plasticity mechanisms and lead to an increase in the number of C1 whisker-responsive neurons in peri-infarct cortex.

Beginning 24 h after stroke, we plucked all whiskers on the contralesional side of the snout, except the C1 whisker (Fig. 4a, b) and continued whisker plucking three times per week until 1 month after stroke. Control animals were subjected to a sham plucking procedure (see "Methods"). We again began by performing ISI to monitor recovery of C1 whisker-evoked activity maps, and extended

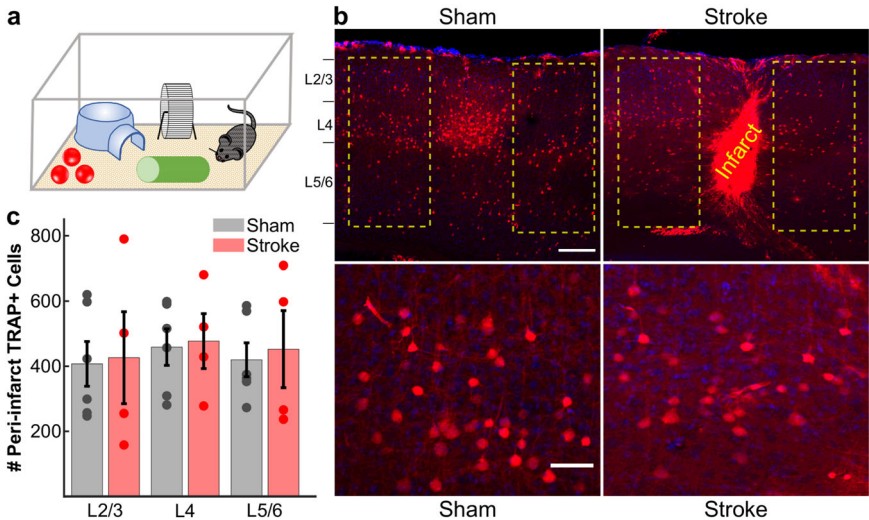

**Fig. 3 TRAP labeling shows no increase in the numbers of C1 whisker-responsive neurons in the peri-infarct cortex 2 months after stroke. a** Schematic of TRAP labeling approach. Two months after stroke targeting the C1 barrel, cFos-CreER^T2:Ai9 mice were subjected to whisker trimming (all whiskers except C1) and then allowed to explore an enriched environment, using only the right-sided C1 whisker for 6 h, immediately following injection of 4-OHT (50 mg/kg in corn oil). **b** Top: representative images (across $n = 6$ sham and $n = 4$ stroke mice) of activity-dependent TRAP labeling in S1BF. Putative C1 whisker-responsive neurons are labeled in red (tdTom); DAPI counterstain. Example regions for quantification are outlined in yellow surrounding either the intact C1 barrel in sham control mice (left panel) or the infarct core in stroke mice (right panel). Note the large density of cFos-expressing (TRAP+) neurons in the C1 barrel in sham control mice. Scale bar = 200 µm. Bottom: high magnification of L4 TRAP-labeled cells surrounding the intact C1 barrel in sham control mice (left panel) or the infarct core in stroke mice (right panel). Scale bar = 50 µm. **c** Quantification of the total number of TRAP+ neurons in the peri-infarct cortex (stroke group, red, $n = 4$) or surround barrels (sham group, gray, $n = 6$), separated by cortical layers. No significant differences were found using a two-way ANOVA for effects of group ($p = 0.736$), layer ($p = 0.828$), or group × layer ($p = 0.995$). Bars show mean ± s.e.m., with data points for individual mice overlaid.

our imaging timepoints to 2 months post-stroke (Fig. 4b–d). The majority of mice did not exhibit a C1 map at 2 months post-stroke, although we did see very small maps in 3/8 mice in the control group and 5/10 mice in the whisker-plucked group (Fig. 4c, d and Supplementary Fig. 5). However, quantification of C1 whisker-evoked ISI signals showed map area and mean signal intensity were both drastically reduced at 5 days after stroke and remained significantly lower than pre-stroke levels up to 2 months post-stroke, even in the whisker-plucked animals (Fig. 4e, f). Stroke size at 5 days post-stroke was similar between groups (Fig. 4g). There was a trend toward an inverse correlation between stroke size and C1 whisker-evoked ISI signal at 2 months post-stroke (Supplementary Fig. 6A), suggesting that the small ISI maps seen in a few animals was perhaps attributable to incomplete ablation of the entire C1 cortical activity map, including portions of barrels immediately adjacent to C1 with significant numbers of C1 whisker-responsive neurons at baseline.

Next, we performed 2P calcium imaging of neuronal responses to C1 whisker stimulation. We imaged multiple fields-of view (FOV) across the peri-infarct region of S1BF (typically 2–4 FOV, with a mean of 32 active neurons/FOV) at each timepoint before and after stroke (Fig. 5a). As expected, the percentage of neurons with stimulus-locked responses to C1 whisker stimulation within the C1 barrel at the pre-stroke baseline was similar between whisker-plucked animals and the control group ($34.3 \pm 2.6\%$ and $34.8 \pm 5.3\%$, respectively; Fig. 5b). Similarly, the percentage of neurons with stimulus-locked responses to C1 whisker stimulation in FOVs adjacent to the C1 barrel was similar between plucked and control groups ($14.1 \pm 1.8\%$ vs. $17.2 \pm 3.1\%$, respectively; Fig. 5c, baseline). Following stroke, we observed a slight decrease in the percentage of C1 stimulus-locked neurons in both control and whisker-plucked groups at 5 and 13 days post-stroke (Fig. 5c), similar to what we had observed in the D3

barrel in our initial cohort of animals (see Fig. 2d, red line). This decrease may be consistent with acute denervation and/or increased inhibition post-stroke[18,19]. By 1–2 months post-stroke, the percentage of neurons in the peri-infarct FOVs with stimulus-locked responses to C1 whisker stimulation had recovered to pre-stroke levels in both groups, without significant differences between the groups (Fig. 5c). A similar trend was observed when normalizing each animal to its pre-stroke baseline percentage of neurons with stimulus-locked responses (Supplementary Fig. 7A); when analysis was restricted to the FOV with the greatest percentage of C1 stimulus-locked neurons at each timepoint (Supplementary Fig. 7B); or when analysis was restricted to the FOV with the smallest percentage of C1 stimulus-locked neurons at baseline (Supplementary Fig. 7C). We also did not find a significant effect of sex on percentage of C1 stimulus-locked neurons over time (Supplementary Fig. 7D), nor did we find a difference when comparing FOV with an ISI map present compared to FOV with no ISI map at 2 months post-stroke (Supplementary Fig. 7E). In other words, the percentage of neurons in peri-infarct cortex that responded to the C1 whisker did not increase above baseline after stroke, even with forced whisker use.

We also considered that the ability of neurons to change their tuning to the C1 whisker after stroke could be affected by stroke size, such that, for instance, larger strokes might hinder plasticity. However, even though animals with larger infarcts at 5 days post-stroke tended to have fewer C1 stimulus-locked neurons at 2 months post-stroke, there was no significant correlation between stroke size and the percentage of stimulus-locked neurons for either cohort of mice (Supplementary Fig. 6B). Overall, the results from plucking experiments again argue against the possibility that new neurons from peri-infarct cortex are recruited to subserve the functions of those lost after stroke.

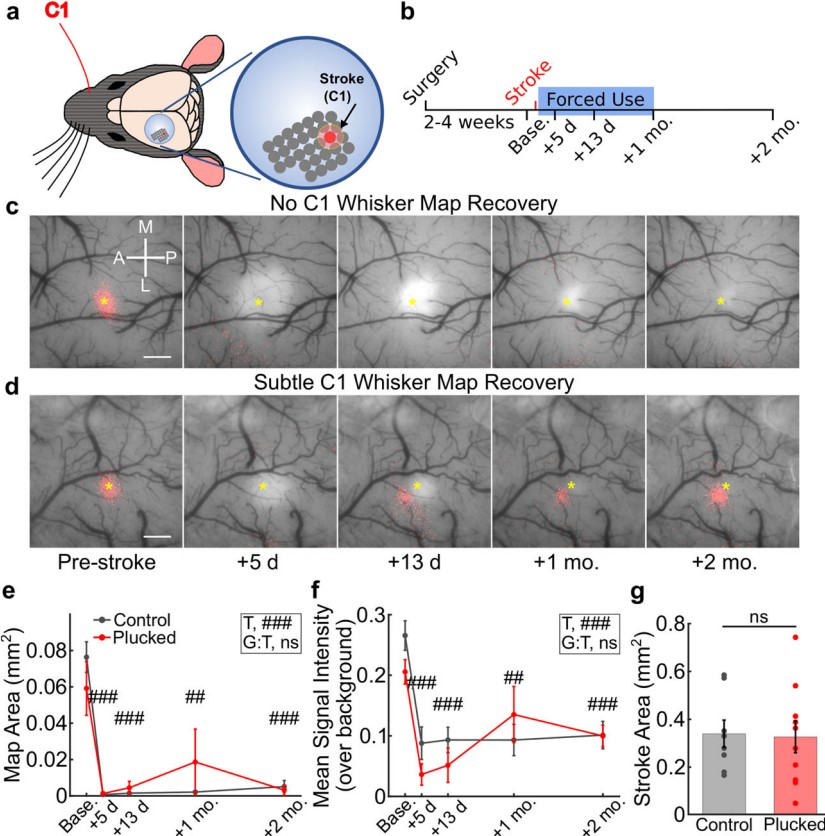

**Fig. 4 Forced use therapy does not trigger macroscopic remapping after C1-targeted stroke. a** Schematic of forced use rehabilitation paradigm, where all right-sided whiskers are plucked except C1. **b** Timeline of surgery, imaging, and forced use whisker plucking. **c**, **d** ISI map of C1 whisker-evoked activity before and after stroke overlaid on photographs of the cranial window in representative mice (with strokes in C1 barrel) that exhibited no recovery of the C1 whisker map (**c**) or subtle partial map recovery (**d**) over 2 months. Scale bar = 0.5 mm. **e** Quantification of C1 whisker-evoked ISI map area size throughout recovery in control (gray, $n = 8$) vs. forced use whisker-plucked (red, $n = 10$) animals. Two-way repeated measures ANOVA, main effects of timepoint (T, ###$p < 0.001$) and group-by-timepoint interaction (G:T, $p = 0.667$). Significance for individual timepoints compared to baseline, corrected for multiple comparisons using the Tukey-Kramer procedure, indicated over corresponding data points (##$p < 0.01$; ###$p < 0.001$). Data in **e** and **f** represent mean ± s. e.m. **f** Quantification of C1 whisker-evoked ISI signals over background throughout recovery in control (gray, $n = 8$) vs. forced use whisker-plucked (red, $n = 10$) animals. Two-way repeated measures ANOVA, main effects of timepoint (T, ###$p < 0.001$) and group-by-timepoint interaction (G:T, $p = 0.213$). Significance for individual timepoints compared to baseline, corrected for multiple comparisons using the Tukey-Kramer procedure, indicated over corresponding data points (##$p < 0.01$; ###$p < 0.001$). **g** Quantification of stroke size at day 5 after stroke from vasculature images in control (gray, $n = 8$) and forced use whisker-plucked (red, $n = 10$) mice. Two-sample, two-tailed $t$ test, $p = 0.888$. Data represent mean ± s.e.m., with individual dots for each mouse overlaid.

**Forced use therapy potentiates sensory-evoked responses in spared whisker-responsive neurons.** Changing the tuning of surviving neurons so that they begin to respond to the C1 whisker after stroke is not the only way that the peri-infarct cortex could contribute to reparative plasticity. For example, we reasoned that the small pool of neurons in surrounding barrels that was already responsive to the C1 whisker before the stroke might begin responding more strongly to C1 whisker stimulation after stroke. To test this, we analyzed sensory-evoked calcium transients for individual neurons in greater detail. Because our stimulation protocol involved 20 sequential whisker stimulations (see "Methods"), we first aligned calcium signals from all individual epochs of whisker stimuli to generate an average sensory-evoked response for each neuron (Fig. 6a). We then calculated both raw Z-scores for the magnitude of the calcium transients and the area-under-the-curve (AUC) as measures of magnitude of neuronal responses after whisker stimulation (see "Methods"). Next, we pooled the average C1 whisker-evoked responses from all neurons with stimulus-locked responses to C1 whisker stimulation for FOVs within either the C1 barrel or surround barrels (SBs) of peri-infarct cortex, for each mouse cohort (whisker-plucked vs.

control). In these stimulus-locked neurons, C1 whisker stimulation elicited large increases (above baseline) in the mean Z-scores of calcium fluorescence intensity (Fig. 6b), whereas neurons classified as non-stimulus-locked did not show any significant sensory-evoked response (Supplementary Fig. 8A). The average C1 whisker-evoked response before stroke for the control cohort was slightly higher in neurons located within the C1 barrel compared to SB neurons (AUC: 7.65 ± 0.97 vs. 5.86 ± 0.92, respectively; Fig. 6b, c). Importantly, there was no significant difference in Z-scores between plucked and control mice at baseline (Fig. 6c).

We next compared sensory-evoked responses of stimulus-locked neurons over time following stroke. In the control group, the mean Z-scores of sensory-evoked responses gradually decreased up to 1 month post-stroke, and only partially recovered by 2 months post-stroke (Fig. 6d, e, black line). In contrast, in the whisker-plucked cohort, the mean Z-score of whisker-evoked responses was only lower 5 days after stroke (3.55 ± 0.62 at +5 days vs. 6.15 ± 1.33 at baseline), and subsequently recovered much more rapidly than in the control group ($p < 0.01$ between groups; Fig. 6d, e, red line). Notably,

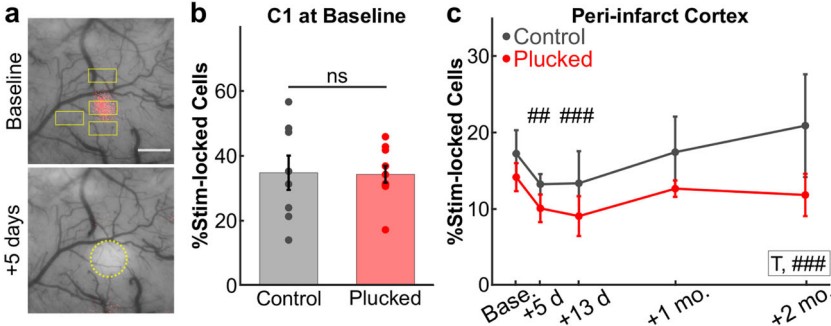

**Fig. 5 Forced use therapy does not lead to an increase in the number of whisker-responsive cells in peri-infarct cortex. a** Representative ISI maps depicting C1 whisker-evoked activity (red) overlaid on a photograph of the cranial window at baseline (top) or 5 days post-stroke (bottom). Yellow rectangles indicate the locations of calcium imaging fields-of-view throughout the peri-infarct region. Stroke at +5 days outlined in yellow circle. **b** Percentage of L2/3 neurons in the C1 barrel with stimulus-locked responses to C1 whisker stimulation at baseline prior to stroke in control (gray, $n = 8$) or forced use whisker-plucked (red, $n = 10$) groups. Two-sample $t$ test, $p = 0.929$. Data represent mean ± s.e.m., with individual dots for each mouse overlaid. **c** Percentage of neurons with stimulus-locked responses to C1 whisker stimulation in the peri-infarct regions throughout recovery in control (gray) and whisker-plucked (red) groups ($n = 8$ and 10 mice, respectively, except $n = 8$ for forced use group at +2 months). GLME binomial model with separate groups, ANOVA for fixed effects of group (G, $p = 0.338$), timepoint (T, $p = 0.208$), and group-by-timepoint interaction (G:T, $p = 0.554$) were not significant. GLME binomial model with pooled groups, ANOVA for fixed effect of timepoint (T, ###$p = <0.001$), with significance for individual coefficients for timepoints (T), corrected using Benjamini and Hochberg's method, indicated over corresponding data points (##$p < 0.01$; ###$p < 0.001$). Data represent mean ± s.e.m.

$Z$-scores in whisker-plucked animals remained slightly higher than baseline from 13 days to 2 months post-stroke, though the difference did not reach significance. Non-stimulus-locked neurons showed minimal sensory-evoked responses at baseline or throughout recovery (Supplementary Fig. 8A), with a trend toward increased responses in the whisker-plucked group, though the magnitude of the difference was very small (Supplementary Fig. 8B). Latency to peak amplitude was largely unchanged over time and not significantly different between groups (Supplementary Fig. 9). Likewise, adaptation, whereby L2/3 neurons in barrel cortex exhibit progressively smaller responses to ongoing bouts of repetitive whisker stimulation[40], was similar between stimulus-locked neurons in the two cohorts of mice (Supplementary Fig. 10).

The larger mean $Z$-scores of sensory-evoked responses we observed for individual neurons could have resulted from responses to individual whisker stimuli that are larger in amplitude, from responses to a greater proportion of the individual whisker stimulations, or from both. Therefore, we next quantified responses from stimulus-locked neurons for each of the 20 distinct whisker stimuli delivered during a given imaging session. Peak $Z$-scores of calcium events differed between groups; after a transient decrease in $Z$-scores in the whisker-plucked group and an increase in control mice (Fig. 6f), by 13 days post-stroke, peak amplitudes had returned to baseline levels in both groups and remained stable through 2 months post-stroke, though they remained slightly higher in the whisker-plucked group (Fig. 6f). Peak amplitudes did not change significantly over time in non-stimulus-locked cells (Supplementary Fig. 8C).

Interestingly, the percentage of stimuli to which individual neurons responded (i.e., had a detectable calcium peak; see "Methods") was significantly different between plucked and control cohorts after stroke (Fig. 6g). Whereas, the percentage of stimuli that neurons responded to in the control group remained below baseline levels until 2 months post-stroke, in the whisker-plucked group it was significantly higher than at baseline (Fig. 6g). The difference between groups was greatest at 13 days after stroke, when, in the whisker-plucked group, there was a ~33% increase in the fraction of stimuli that neurons responded to compared to baseline (30.8 ± 2.6% of stimulation

epochs vs. 23.2 ± 1.9% at baseline), while the control group showed a reduction in this value compared to baseline. A similar trend for increased likelihood of neuronal activity in animals from the whisker-plucked cohort was seen in non-stimulus-locked cells, though the magnitude of the difference was much smaller (Supplementary Fig. 8D). We also observed an increase in the frequency of spontaneous calcium transients (i.e., in the absence of whisker stimulation) in the whisker-plucked cohort (Supplementary Fig. 11).

## Discussion

A major gap in our understanding of functional recovery after stroke has been how specific changes in neuronal activity mediate such recovery. The prevailing dogma is that circuits remap by recruiting surviving neurons to assume new functions, namely those previously encoded by neurons that died[4,49]. Although this model is attractive and makes intuitive sense, the evidence to support it has been largely indirect and unconvincing. Here, we sought to directly test the remapping hypothesis by recording neural activity in peri-infarct cortex before and after stroke. Using three different in vivo approaches (ISI, 2P calcium imaging, and TRAP), we find no evidence of remapping of lost functionalities. That is, we could not identify any increase in the population of C1 whisker-responsive neurons, which would be expected if surviving neurons in peri-infarct cortex were multitasking to assume the role of the dead neurons. On the contrary, we find that the proportion of whisker-responsive cells decreases acutely after stroke, before returning only to baseline levels over 1–2 months (Fig. 7). However, in rehabilitated animals (using whisker plucking as a means of forced use therapy), we find significant increases in the reliability of whisker-evoked responses in circuits that were already responding to that whisker before the stroke (Fig. 7c, middle panel). Our findings are significant because they put into question the long-held remapping model of stroke recovery that has influenced stroke research for decades.

Much of the remapping hypothesis for stroke recovery is predicated on human brain mapping studies showing different patterns of activation after stroke[9], but which ultimately came short of supporting the remapping theory because of their inherent variability. The imaging methods employed also suffer from several technical and practical limitations. For example, functional MRI

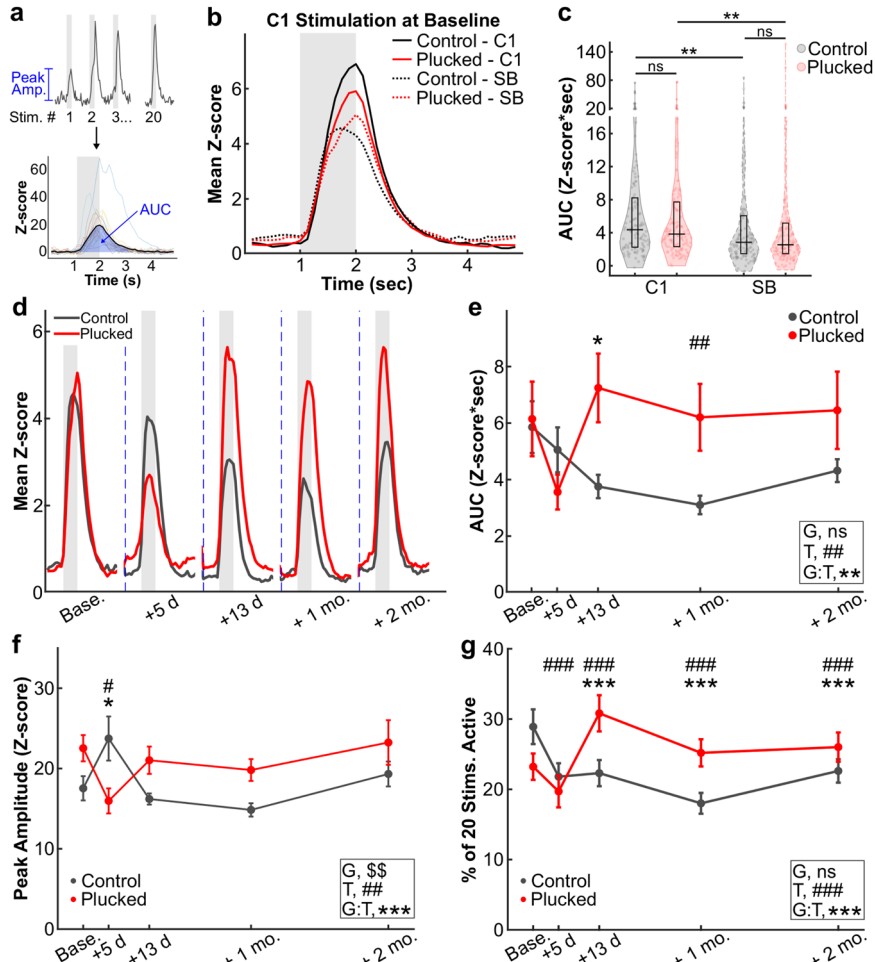

**Fig. 6 Forced use therapy after stroke increases the reliability of sensory-evoked responses of neurons in peri-infarct cortex. a** Analysis pipeline. Top: neuronal responses to 20 sequential whisker deflections at 10 Hz were aligned to the onset of the stimulus epoch (gray shading) to generate a mean stimulus-evoked trace per cell. Bottom: individual trials (thin lines) and mean trace (thick line). Peak amplitudes for every stimulus were calculated from individual calcium transients (top), while the area-under-the-curve (AUC) was calculated from the mean stimulus-evoked trace (bottom). **b** Mean stimulus-evoked response at baseline for all stimulus-locked neurons in control (black) vs. forced use whisker-plucked (red) mice, comparing neurons located in the C1 barrel (C1—solid lines; $n = 121$ neurons from 8 control mice and 145 neurons from 10 plucked mice) to those in surround barrels (SB—dashed lines; $n = 128$ and 146 neurons from the same mice, respectively). Same $n$ applies for **c**. **c** Quantification of the AUC from mean stimulus-evoked responses for all stimulus-locked neurons at baseline. Control group in gray, whisker-plucked group in red. Each dot represents one neuron, horizontal black lines show group medians, and black rectangles show 25–75th percentiles. Kruskal–Wallis test with Tukey-Kramer correction for multiple comparisons, $**p < 0.01$. **d** Mean stimulus-evoked response from all cells in peri-infarct regions from control (black) or plucked (red) mice over time following stroke. Number of cells ($n$)/number of mice ($N$) for control and plucked groups: baseline—128/8 and 146/10; day 5—89/8 and 84/10; day 13—165/8 and 110/9; 1 month—148/8 and 155/10; 2 months—128/8 and 119/8. Same $n$ applies to **e**. **e** Quantification of the AUC from the mean stimulus-evoked responses from control or plucked mice shown in **d**. LME model, ANOVA for fixed effects of group (G, $p = 0.646$), timepoint (T, $##p = 0.007$), and group-by-timepoint interaction (G:T, $**p = 0.003$). Significance for individual coefficients for timepoint (T) or group-by-timepoint interaction (G:T), corrected using Benjamini and Hochberg's (B & H) method, are indicated over corresponding data points ($*p < 0.05$; $##p < 0.01$). Data in **e**–**g** represent mean ± s.e.m. **f** Quantification of peak amplitude of stimulus-evoked responses in stimulus-locked neurons over time following stroke, in control vs. plucked (red) groups. Number of cells ($n$)/number of mice ($N$) for control and plucked groups: baseline—119/8 and 135/10; day 5—85/8 and 75/10; day 13—152/8 and 104/9; 1 month—135/8 and 146/10; 2 months—123/8 and 112/8. LME model, ANOVA for fixed effects of group (G, $$$p = 0.003$), timepoint (T, $##p = 0.002$) and group-by-timepoint interaction (G:T, $***p < 0.001$). Significance for individual coefficients for timepoint (T) or group-by-timepoint interaction (G:T), corrected using B & H method, are indicated over corresponding data points ($*$ or $#p < 0.05$). **g** Quantification of the percentage of whisker stimuli that stimulus-locked neurons respond to over time following stroke, in control vs. plucked groups. $n/N$ for control and plucked groups: baseline—128/8 and 146/10; day 5—89/8 and 84/10; day 13—165/8 and 110/9; 1 month—148/8 and 155/10; 2 months—128/8 and 119/8. GLME binomial model, ANOVA for fixed effects of group (G, $p = 0.054$), timepoint (T, $###p < 0.001$) and group-by-timepoint interaction (G:T, $***p < 0.001$). Significance for individual coefficients for timepoint (T) or group-by-timepoint interaction (G:T), corrected using B & H method, are indicated over corresponding data points ($***$ or $###p < 0.001$).

and fluorodeoxyglucose PET rely on surrogate markers of neuronal activity, such as blood flow, oxygen content, or metabolism (all of which are altered by stroke), and they lack the temporal and spatial resolution necessary for recording activity in single neurons over time[50]. Most importantly, the human studies are missing a pre-

stroke baseline imaging session, which is critical to interpret whether remapping occurred or whether observed changes represent baseline variability across subjects. Our goal was to overcome these limitations by recording from neuronal populations longitudinally before and after stroke, using a model that led to

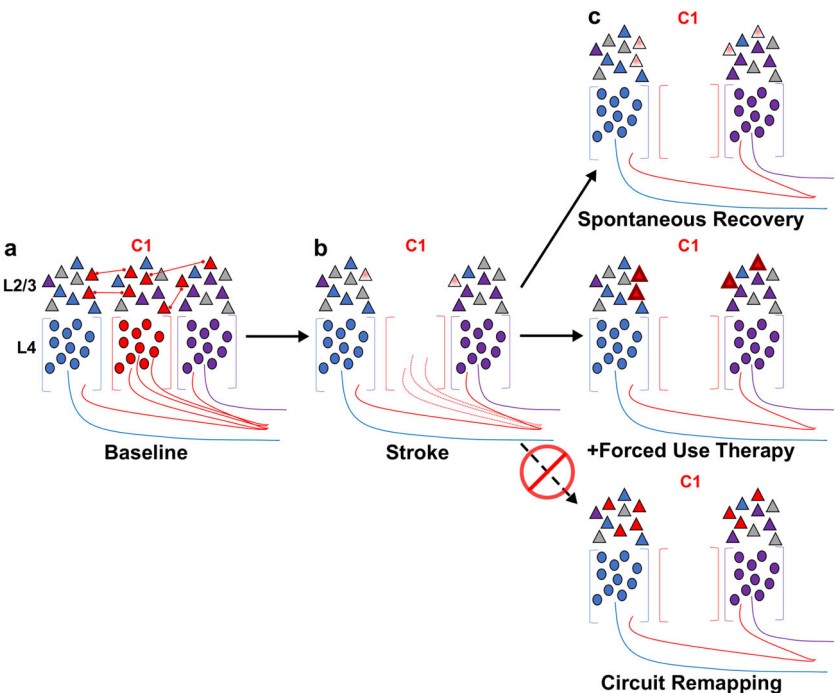

**Fig. 7 Models of L2/3 whisker somatosensory circuit changes post-stroke. a** Distribution of whisker-responsive L2/3 pyramidal neurons (triangles) in three adjacent barrels of the S1BF at baseline. Colors denote responsivity to a particular whisker (red = C1). Thalamocortical axons are depicted innervating L4 stellate neurons (circles) within barrels, with barrel septa pictured as brackets. Intracortical connections between similarly tuned L2/3 pyramidal neurons are depicted as well. Note that within a given barrel, L2/3 neurons are tuned to both the principal and surround whiskers. **b** After stroke targeting a single barrel, all neurons within the barrel are destroyed. The proportion of surround barrel neurons that are tuned to the C1 whisker (corresponding to the infarcted barrel) is decreased after stroke and their sensory-evoked responses are reduced (paler shading). **c** Spontaneous recovery (top panel) results in restoration of the proportion of surround barrel neurons tuned to the C1 whisker destroyed by stroke, but their sensory-evoked responses remain depressed, and there is no retuning of spared neurons to replace neurons lost to stroke. Forced use therapy (middle panel) after stroke restores and potentiates sensory-evoked responses in surround whisker neurons, but does not result in true circuit remapping with recruitment of new C1 whisker-responsive neurons (bottom panel).

consistent and reproducible infarcts of relatively small size. We reasoned the S1BF would be an ideal location for these studies because it exhibits a unique somatotopic organization at a macroscopic level: each individual whisker is primarily mapped to a single cortical barrel[28], and yet, within individual barrels, many neurons are tuned to their corresponding principal whisker, while others respond to surround whiskers[31]. Importantly, tuning is highly dynamic in the intact S1BF, with neurons capable of modifying their selectivity in response to changes in sensory experience. For example, after whisker plucking, L2/3 neurons rapidly shift their tuning away from deprived whiskers and toward spared whiskers[32]. However, whether lesioning the central whisker representation in the cortex, i.e., its corresponding barrel, triggers similar changes in tuning of neurons in surrounding barrels has never been documented.

The only way to settle this debate was to conduct in vivo calcium imaging though cranial windows over several weeks after stroke, while stimulating single whiskers. After a stroke targeting the C1 barrel, we expected that spared neurons in SBs would change their tuning, resulting in greater numbers of C1 whisker-responsive neurons, but instead we found the opposite result. We also did not find an increase in putative C1 whisker-responsive neurons using activity-dependent labeling in TRAP mice, nor in our separate forced use therapy cohort. Overall, our data provide robust evidence against the remapping hypothesis for stroke recovery, as we could not identify new recruitment of spared neurons to subsume lost functionalities.

Other studies have identified molecular and structural changes in animal models that have been interpreted as providing the substrate for circuit remapping. These include transcriptomic changes involving plasticity-related genes[51], sprouting of axons[52], and dendritic spine turnover[53–55]. However, definitive evidence showing that these changes actually result in spared neurons assuming the functions of neurons lost to stroke remains lacking. Transcriptional changes could support structural plasticity, but this could just as easily reflect circuit plasticity to augment existing connectivity, rather than true functional remapping. This highlights the need for additional studies to explore the consequences of structural plasticity on the ultimate functionality of neurons after stroke. Perhaps the best evidence to date for the remapping theory was the pioneering study of Winship and Murphy in 2008, in which they reported that a small population of limb somatosensory neurons exhibited broader tuning 1–2 months post-stroke[14]. There are several important differences between that study and ours that may account for the different results, including the stroke location and brain region imaged (forelimb/hindlimb S1 vs. S1BF), and the type of imaging (acute recordings with Oregon Green BAPTA-1 in different mice before and after stroke vs. chronic imaging of GCaMP6s expressing neurons in the same mice over time). Although we are not aware of other studies recording sensory-evoked activity of cortical neurons in the peri-infarct cortex, the activity of forelimb thalamocortical axons is persistently impaired post-stroke[56], a result which is generally in line with our findings. However, it remains possible that mechanisms of functional remapping after stroke could differ across cortical areas. For instance, there have been reports of functional remapping by ISI in the forelimb representation in S1 (refs. [11,57]), though importantly these studies

did not examine functional remapping at the level of individual neurons. As such, additional studies are required to determine if our findings in S1BF generalize to other cortical regions.

One could argue that we failed to detect remapping because we were imaging in the wrong location, by surveying regions surrounding the C1 barrel rather than more distant locations. However, this is unlikely because the tuning of L2/3 neurons in adjacent barrels is known to be promiscuous[31], and plasticity triggered by whisker plucking in the intact S1BF preferentially recruits adjacent barrels[26,32]. Subcortical plasticity may occur, especially at the level of the thalamus, but this is probably insufficient to completely restore sensory function; it may help with whisker detection, but for discrimination, S1BF is essential[58,59]. It is also likely that stroke in the S1BF, and the resulting changes in sensory-evoked responses that we have uncovered, induce alterations in the activity of downstream circuits on a more macroscopic scale, for example, in regions, such as secondary somatosensory cortex or whisker motor cortex. Such macroscopic circuit changes have been interpreted as evidence of remapping in the past, as discussed above. However, our aim in this study was to test the specific hypothesis that observed macroscopic activity changes are driven by retuning of functionality of individual neurons, and our data strongly suggest against this. Going forward, it may be helpful in the field to more precisely define the term remapping, as it is sometimes used to describe plasticity changes after stroke at synaptic, neuronal, circuit, and brain-wide scales. In the context of our paper, we considered remapping to mean that neurons that survived the lesion (in this case stroke) change their tuning to assume "de novo" the functions of neurons that were lost.

Another potential criticism is that the small lesions we created may not produce a sufficient functional deficit to drive remapping. However, other studies have shown that ablation of single barrels does lead to functional impairment[60], and we found that larger strokes were actually associated with a trend toward even fewer whisker-responsive cells at 2 months. A different concern related to the size and location of the stroke is that it might have extended to the dysgranular zone next to the barrel field, an area with long-range projections to other regions of somatosensory cortex[61]. It is conceivable that extensive damage to this area could impact post-stroke plasticity or remapping. We think such damage in our study is rather unlikely because we show that strokes targeted to C1 did not extend beyond the neighboring barrels, and therefore did not damage the dysgranular zone in any significant way. It is possible that our calcium imaging approach was not sensitive enough to detect remapping. Yet, we were able to detect significant increases in the reliability of how surviving neurons respond to their principal whisker, and that whisker plucking further enhanced such changes. Thus, plasticity does occur after stroke, but it does not appear to involve spared neurons taking on new functions. It is also unlikely that remapping is mediated by a small number of neurons scattered throughout peri-infarct cortex because we probed hundreds of whisker-responsive neurons in vivo and found only one animal (out of 18) with a significantly higher percentage of neurons that responded to the C1 whisker after stroke, compared to baseline. Finally, our imaging was performed under light anesthesia, so it is possible that remapped neurons would have been observed in awake-behaving animals performing active whisking tasks. However, we did not see any evidence of remapping in our experiments using activity-dependent labeling (TRAP) either, so we think this is unlikely.

One can only speculate about why having neurons multitask after cortical injury (an essential aspect of the remapping theory) may not pose an evolutionary advantage. Even if remapping were to occur, it is unclear what the functional advantages of broadened tuning of neurons post-stroke might be. Sparse firing of

cortical neurons to specific stimuli underlies efficient sensory processing in S1 (ref. [62]). It is conceivable that, as spared neurons assume additional roles post-stroke, this might degrade sensory processing. In line with this hypothesis, we and others found that even a small increase in tuning width of neurons in the visual cortex leads to impaired visual discrimination in mice[63,64]. Furthermore, the fact that whisker plucking (which triggers tuning changes in healthy S1 cortex) did not promote circuit remapping after stroke suggests that mechanisms exist within the cortex that hinder remapping. This interpretation is supported by prior studies showing impaired plasticity in peri-infarct cortex, including the absence of whisker trimming-induced expansion of cortical maps[21,22], and loss of monocular deprivation-induced ocular dominance shifts[20]. The mechanisms underlying this curtailment in plasticity are not known, but may include alterations in the excitation/inhibition balance, such as increased tonic inhibition[18,19]. Indeed, C1 whisker-evoked responses were reduced after stroke, which is consistent with such an increase in inhibition. Given that whisker plucking is known to induce circuit remapping via disinhibition in the intact S1BF, we favor a model in which increased inhibition limits circuit remapping after stroke, though further studies will be necessary to clarify these mechanisms.

Although forced use therapy did not induce circuit remapping after stroke, it did accelerate the recovery of whisker-evoked calcium transients in spared C1-responsive neurons and increased the reliability of their responses to individual stimuli (Fig. 6g). We believe we have uncovered a circuit basis for how rehabilitation via forced use therapy might promote recovery after stroke. Specifically, we hypothesize that forced use rehabilitation is effective by boosting responses of spared neurons already serving a particular function before stroke, rather than by changing the selectivity of other neurons to acquire new functions. Of course, forced use therapy in humans is typically employed for motor deficits (by immobilizing a spared limb contralateral to the paretic limb). Still, we believe our results do have significant implications for physical rehabilitation strategies. In order to promote the true circuit remapping (Fig. 7c, bottom panel), rehabilitation may need to be combined with genetic[57], pharmacologic[18], or other approaches[56,65] to circumvent mechanisms limiting remapping post-stroke.

In conclusion, while the lack of remapping after stroke we describe is unexpected and perhaps counterintuitive to some, it cannot be ignored. Functional recovery, if and when it occurs after stroke, is not mediated by surviving neurons assuming new roles (multitasking), but may instead involve allocating resources to potentiate pre-existing circuits, at least in S1 barrel cortex. Additional studies will be needed to explore functional remapping in other areas of cortex. However, until it is unequivocally demonstrated that neurons can consistently undertake the roles of neighbors that were lost after cortical lesions, we believe that the established hypothesis for functional circuit remapping after stroke should be viewed with skepticism.

## Methods

**Materials**. All chemicals were obtained from Sigma Aldrich unless otherwise noted.

**Experimental animals**. All experiments followed the U.S. National Institutes of Health guidelines for animal research, under an animal use protocol approved by the Chancellor's Animal Research Committee and Office for Animal Research Oversight at the University of California, Los Angeles (#2005-145). Both male and female mice were used, beginning at 6–10-weeks-old at the time of cranial window surgery. All animals were housed in a vivarium with a 12 h light/dark cycle, temperature ~23 °C (range 20–26 °C), and humidity 30–70%. For in vivo imaging, we used transgenic *Thy1*-GCaMP6s mice (GP4.3, JAX line 024275)[66]. For activity-dependent labeling, we used the TRAP approach[42], crossing *cFos*-CreER[T2] mice (JAX line 021882) with the Ai9 Cre-dependent tdTomato reporter line (JAX line 007909). One *Scnn1a*-Tg3-Cre:Ai162 mouse (JAX line 009613 crossed with JAX

line 031562) was used for labeling of barrels in S1BF and tangential sectioning. All transgenic lines were maintained on a C57BL/J6 background.

**Cranial window surgery**. We implanted chronic glass-covered cranial windows in 6–10-week-old mice, as described previously[67,68]. Animals were deeply anesthetized using 5% isoflurane for induction followed by maintenance with 1.5–2% isoflurane. A circular craniotomy, ~4–5 mm in diameter, was made using a pneumatic dental drill over the primary somatosensory cortex, centered ~3 mm lateral to the midline, and ~2 mm caudal to Bregma. A sterile glass coverslip (#1; Electron Microscopy Sciences) was placed over the craniotomy and glued to the skull with cyanoacrylate glue (Krazy Glue). Dental acrylic (OrthoJet, Lang Dental) was then applied throughout the exposed skull surface around the edges of the coverslip. To later secure the mouse onto the microscope stage, a small titanium bar was embedded in the dental acrylic. Carprofen (5 mg/kg, i.p., Zoetis) and dexamethasone (0.2 mg/kg, i.p., Vet One) were provided for pain relief and mitigation of edema on the day of surgery and daily for the next 48 h. Mice were allowed to recover from the surgery for 2–4 weeks before the first imaging session.

**Intrinsic signal imaging**. Whisker-evoked sensory activity maps were generated using ISI[69]. Animals were sedated with chlorprothixene (~3 mg/kg, i.p.) and lightly anesthetized with ~0.5–0.7% isoflurane, were head-fixed to the stage of a custom-built tandem-lens (135 and 50 mm focal lengths, Nikon) macroscope. The cortical surface was illuminated by green LEDs (535 nm) to visualize the superficial vasculature. The macroscope was then focused ~300 μm below the cortical surface and red LEDs (630 nm) were used to record intrinsic signals, with frames collected at 30 Hz 0.9 s before and 1.5 s after stimulation, using a fast CCD camera (Teledyne Dalsa Pantera 1M60), a frame grabber (64 Xcelera-CL PX4, Dalsa), and custom routines written in MATLAB (version 2009a). Thirty trials separated by 20 s were conducted for each imaging session. Whisker stimuli (100 Hz, 1.5 s duration) were delivered by carefully affixing an individual whisker to a glass capillary tube coupled to a ceramic piezoelectric bending actuator (Physik Instrumente). Evoked signals were quantified in two ways. To quantify map area, standardized Z-scores of raw change in reflectance values ($\Delta R/R$) for ISI images were calculated in MATLAB (version 2020a). A threshold of $Z < -3$ was then applied (see Supplementary Fig. 2B–E) and the area of thresholded pixels was calculated. For the experiment comparing control and forced use cohorts (Fig. 4e), we could not locate the images for raw $\Delta R/R$ values for two mice that were collected >2 years previously; therefore, we analyzed instead the intensity scaled (as a percentage of maximum) images. ISI signal intensity values were calculated using ImageJ (version 2.1.0) as the mean pixel intensity of the evoked whisker map normalized to the background signal outside of the map. If a map could not be visualized (for instance, following stroke), the mean pixel intensity of the area adjacent to the stroke was normalized to the background signal. For representative images of whisker-evoked ISI maps (Figs. 1d, 4c, d, and 5a), raw ISI signals were thresholded using $Z < -3$ to create a binary mask. The mask was then overlaid on the cortical vasculature image.

**Photothrombotic stroke**. Animals were deeply anesthetized as above, and core body temperature was maintained at 37 °C using a homeothermic blanket system (Harvard Apparatus). Rose Bengal dye (120 mg/kg of mouse body weight, diluted in sterile saline, i.p.), was injected 10 min prior to head-fixing the mouse on the stage of a custom-built 2P microscope. The beam of a green laser (532 nm, ~2 mW intensity at the sample; Laserlands 1875-532D) was aligned through the optical path of the microscope and scanned across an ~0.5 × 0.5 mm region of the cortical surface corresponding to the C1 barrel (previously identified using ISI) for 10 min. Animals were allowed to recover on a heated water-recirculating blanket in the dark for 1 h before being returned to the home cage. Proper targeting of the stroke to the C1 barrel was confirmed using ISI 5 days after stroke, and animals in which the stroke was mistargeted were excluded from further analysis. For the sham control group, animals were injected with an equivalent volume of saline, but otherwise treated the same, including the green laser scanning across the C1 barrel.

**Cresyl violet staining**. Five days after stroke animals were transcardially perfused with 4% paraformaldehyde in phosphate-buffered saline (PBS). Brains were removed and 50 μm thick coronal sections were cut using a vibrating microtome (Leica VT 1000). Sections were sequentially mounted, dried, and placed into 1:1 ethanol:chloroform overnight. The following day, sections were rehydrated through 100 and 95% ethanol, rinsed in dH₂O, and then stained in 0.1% cresyl violet solution (warmed to 37 °C) for 10 min. After staining, sections were rinsed in dH₂O, differentiated in 95% ethanol for 10 min, dehydrated in 100% ethanol, and cleared in xylene. Sections were imaged on an upright fluorescent microscope (Zeiss Axio Imager 2 with Zen Pro 2.5 software), using a 10× objective.

**Tangential sectioning**. A Scnn1a-Tg3-Cre:Ai162 mouse was perfused with 4% paraformaldehyde 5 days after C1-targeted PT stroke. The cortex was then dissected and flattened between two glass slides separated by 1 mm thick spacers. Tangential sections, 50 μm thick, were cut, mounted in Fluoromount-G with DAPI (ThermoFisher), and imaged on an upright fluorescent microscope (Zeiss Axio Imager 2 with Zen Pro 2.5 software), using 5× and 10× objectives.

**Whisker trimming/plucking**. After animals were appropriately anesthetized (as described above), all whiskers on the right side of the snout, except those undergoing stimulation (C1 and/or D3), were trimmed using a fine scissors to a length of ~5 mm on the morning prior to imaging. For the forced use whisker plucking group, all whiskers on the right side of the face except C1 were entirely plucked from the follicle starting 24 h after stroke. Additional plucking was performed three times weekly to remove any whisker regrowth. After 1 month, whiskers were allowed to regrow. For sham plucking, mice were similarly anesthetized, and we handled their whiskers with forceps but did not actually remove the whisker.

**Two-photon calcium imaging**. In vivo calcium imaging was performed using a custom-built 2P microscope with galvanometric scanners (Cambridge Technology), a Chameleon Ultra II Ti:sapphire laser (Coherent), a 20× objective (0.95 NA, Olympus), and ScanImage software (version 3.8, Vidrio Technologies)[40]. Data for 3/18 mice in the forced use experiment (one control animal at 1 month, one plucked animal at 1 month, and one control animal at 2 months post-stroke) were collected on a galvo-resonant microscope (Neurolabware). Mice were lightly sedated with chlorprothixene (~3 mg/kg, i.p.) and isoflurane (0.7–0.9%) and kept warm with a heating blanket. Stimulation of the C1 or D3 whiskers (20 stimuli, 1 s duration at 10 Hz, with a 3 s interstimulus interval) was delivered with a piezoelectric actuator, as above. Whole-field images were acquired with bidirectional scanning at ~7.8 Hz (1024 × 128 pixels, downsampled to 256 × 128 pixels). On average, we recorded 31.8 active neurons per FOV. In total, we recorded from 13,991 neurons in 33 mice. For each imaging FOV, ~100 s of spontaneous activity data was collected prior to sensory-evoked activity. In our first cohort of mice comparing sham stroke to C1 barrel-targeted stroke, FOV were selected based on the location of the C1 and D3 ISI activity maps. In our second cohort of mice, comparing forced use whisker plucking to control, 2–4 FOV adjacent to the C1 barrel were imaged (on average, the distance from the center of the C1 barrel FOV to the center of the SB FOV was 473 ± 11 μm). After stroke, we could not image some FOV due to damage from the stroke or stroke-induced shifts in vasculature. In these cases, the FOV closest to the original FOV was chosen, typically immediately adjacent to the edge of the infarct.

Neuropil-corrected change in fluorescence traces ($\Delta F/F$) of neuronal calcium transients were extracted using custom-written semiautomated MATLAB routines[40,70,71]. The vast majority of movies (>90%) did not require motion correction, but for those that did, X–Y drift was corrected using either a frame-by-frame, hidden Markov model-based registration routine[72] or the motion correction module from EZcalcium[73] based on NoRMCorre nonrigid template matching. Modified Z-scores of $\Delta F/F$ values were then calculated for each neuron and used for all subsequent analyses, according to the equation:

$$Z(t) = \frac{[F(t) - \text{mean(quietest period)}]}{\text{Standard deviation(quietest period)}} \quad (1)$$

where the "quietest period" is the 10 s period with the lowest variation in $\Delta F/F$. To determine whether individual neurons showed stimulus-locked responses to whisker stimulations, we used a probabilistic bootstrapping method to correlate calcium transients with epochs of stimulation[40]. We generated 10,000 scrambles of the Z-score activity vector for each cell (preserving all calcium activity epochs with Z-scores above three for at least four consecutive frames), and calculated the correlation between these scrambled vectors and the whisker stimulus signal vector. We then calculated the correlation between the actual Z-score activity vector for the cell and the whisker stimulus time vector, and compared this value with the distribution of correlations for our 10,000 scrambled activity vectors. Cells with correlation values in the top 1% of the scrambled distribution were considered stimulus-locked.

For spontaneous activity, amplitude and frequency of calcium transients were detected using the PeakFinder script (Mathworks File Exchange, version 2.0.2.0) in MATLAB (version 2020a). We also calculated the total AUC of the entire ~100 s Z-score trace. Sensory-evoked activity from cells was quantified in several ways using custom scripts in MATLAB (version 2020a)[74]. Cells were separated into groups based on whether they exhibited stimulus-locked responses to whisker stimulation (stimulus-locked or non-stimulus-locked). Modified Z-scores for each of the 20 individual whisker stimuli were aligned to the onset of the stimulus and a mean stimulus-evoked trace was calculated for each cell. The AUC of the mean stimulus-evoked trace was quantified as the trapezoidal integral of the modified Z-score vector over 2 s after stimulus onset. In this way, the AUC of the mean stimulus-evoked trace captures the stimulus-evoked activity of a cell across all 20 stimuli. For each neuron we also used the PeakFinder script to identify the presence (or absence) of an evoked response, the peak amplitude of that response, and the latency to peak for each of the 20 individual whisker stimuli delivered in a given imaging session. For peak amplitude and latency, the mean response was calculated only for stimulus epochs with a detected peak; stimulus epochs without a detected peak were ignored. Because many neurons show decreases in the amplitude of their calcium signals with ongoing bouts of whisker stimulation[40], we also calculated adaptation indices across the 20 stimuli as:

$$\frac{\left[(\text{AUC during stimulations } 1-5) - (\text{AUC during stimulations } 16-20)\right]}{\left[(\text{AUC during stimulations } 1-5) + (\text{AUC during stimulations } 16-20)\right]} \quad (2)$$

**Activity-dependent labeling**. We used a transgenic mouse line, in which the Cre recombinase is knocked in downstream of the promoter for the immediate early

gene *cFos*, to enable activity-dependent labeling of neurons during a defined epoch[42]. This approach has been termed targeted recombination in active populations or TRAP. We crossed these *cFos*-CreER$^{T2}$ mice with the Cre-dependent reporter line Ai9. TRAP mice underwent cranial window implantation and stroke (or sham stroke) of the C1 barrel. ISI imaging was used first to locate the C1 barrel and then again 1 week after stroke to confirm proper targeting. Two months after stroke, mice were brought to the behavior testing room and all whiskers were trimmed flush to the face, except for the right-sided C1 whisker. The following day, mice were injected with 4-hydroxytamoxifen (50 mg/kg in corn oil, i.p.)[42]. Following injection, mice were placed in pairs in an environmental enrichment cage (13–1/4 in. × 16–19/32 in. × 7–3/8 in.; 1800 Mouse Cage, Lab Products), containing tunnels, a running wheel, and other toys of various shapes and sizes, in the dark for 6 h. Mice were then returned to their home cages for 3–7 days, followed by transcardial perfusion with 4% paraformaldehyde in PBS.

Brains were removed and the left hemisphere was carefully dissected and cut into 60 µm thick coronal sections, using a vibrating microtome (Leica VT 1000). Sections were sequentially mounted using Fluoromount-G with DAPI (ThermoFisher). The section containing the infarcted or intact (for sham animals) C1 barrel was manually identified; next four sections anterior and two sections posterior, totaling a span of ~360 µm in the anterior/posterior direction, were imaged on an upright fluorescent microscope (Zeiss Axio Imager 2 with Zen Pro 2.5 software), using a 10× objective, Apotome.2 optical sectioning module, and DAPI or DsRed filter sets. Images were then randomized and blinded for quantification of tdTomato-labeled putative C1 whisker-responsive cells in the peri-infarct cortex. The S1BF was first identified on each DAPI-labeled image and manually annotated, using ImageJ (version 2.1.0) and the mouse Allen Brain atlas as a reference. The region-of-interest corresponding to the S1BF was then transferred to the tdTomato images and all tdTomato-positive cells were manually counted, by cortical layer, within the S1BF region. Labeled cells within the ischemic core or the C1 barrel were excluded from analysis of peri-infarct labeled cells. Cells from one coronal section from one sham animal could not be counted due to tissue folding. This data point was imputed with the mean value for the corresponding sections from the remaining sham animals. Neither replacing this value with twice nor half the mean value for corresponding sections, nor excluding data from this animal entirely had any effect on the statistical findings.

**Statistics and reproducibility**. All data are shown as mean ± standard error of the mean, unless otherwise stated. Sample sizes, indicated in figure legends, were not based on a priori power calculations, but are consistent with other studies in the field using similar techniques, including our own[40,55,63,69]. We did not repeat experiments on additional independent cohorts of animals. Cresyl violet staining was performed on two animals to confirm stroke depth. Tangential sections were cut from one *Scnn1a*-Tg3-Cre:Ai162 mouse to confirm that strokes did not disrupt the dysgranular zone surrounding the S1BF. Statistical analyses were performed in MATLAB (version 2020a). Data were tested for normality using the Lilliefors test and analyzed using parametric or nonparametric statistical tests, as indicated in the figure legends. For the percentage of whisker-responsive cells (Figs. 2c, d and 4c), we used generalized linear mixed-effects models with binomial distribution, according to the model "responsive ROIs ~ 1 + group + timepoint + timepoint × group + (1 | mouseID)", with each animal serving as one data point. In the analysis of percentage of whisker-responsive cells comparing control to whisker-plucked groups (Fig. 4c), we found no effects of group or group-by-time interaction, so data were pooled and reanalyzed just for effects of time according to the model "responsive ROIs ~ 1 + timepoint + (1 | mouseID)".

For analysis of AUC, peak amplitude, and peak latency of sensory-evoked calcium events (Figs. 5e, f and Supplementary Fig. 6), linear mixed-effects models on data from individual neurons were used, according to the formula "*y* ~ group + timepoint + group × timepoint + (1 | mouseID)". Residuals from raw data were non-normal, so data was log-transformed for further statistical analysis. To facilitate log transformation, a constant value was added to any datasets with negative values, such that the minimum value for the dataset was equal to 0.1. For analysis of the percentage of stimuli for which a neuron was active, a generalized linear mixed-effects model was used with binomial distribution. For all linear mixed-effects models, *p* values for individual coefficients from the model were corrected for multiple hypothesis testing using the Benjamini and Hochberg procedure[75]. ANOVA was used to test for overall effects of group, time, and group-by-time interactions. In the figures, results are presented with p values for overall effects of group (G), timepoint (T), and group-by-timepoint interactions (G:T) shown as an inset on respective graphs, with *p* values for individual coefficients depicted adjacent to the respective data points.

This is the first longitudinal in vivo 2P calcium imaging study to record neuronal activity for weeks before and after stroke. In some animals, the cranial window did not remain optically transparent for every single imaging timepoint. As a result, data for some timepoints could not be collected. For the analysis of the percentage of neurons with stimulus-locked response to C1 and D3 whisker stimulation, comparing sham vs. stroke groups, the following data points were missing: one mouse in the stroke group at 13 days post-stroke and four mice in each group at 1 month post-stroke. For the analysis of sensory-evoked responses in control vs. forced use whisker-plucked groups, one mouse in the plucked group

had no neurons with stimulus-locked responses to C1 whisker stimulation at +13 days post-stroke, and two mice in the plucked group were not imaged at 2 months post-stroke.

For correlation between stroke size and percentage of stimulus-locked cells or ISI signal, data were fit using a simple linear regression model. Values for $r^2$ (adjusted for number of coefficients) and *p* are presented on the graphs for pooled data. Analysis was also conducted on each group separately and no significant correlation was seen for either group individually.

**Reporting summary**. Further information on research design is available in the Nature Research Reporting Summary linked to this article.

## Data availability
The data generated and analyzed for this study are available from the corresponding author upon reasonable request. Source data are provided with this paper.

## Code availability
MATLAB code used for ROI selection (DC_Calcium), analysis of sensory-evoked responses (DC_Calcium Analysis Scripts), and motion correction (EZCalcium) are all available for download from Github.

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

## Acknowledgements

This work was supported by R01NS076942-05 (to C.P.-C.) and R25-NS065723 (to W.A.Z.) from NIH-NINDS, as well as NRTS-2199 (to W.A.Z.) from the American Academy of Neurology. We thank Drs. Cynthia He and Daniel Cantu for MATLAB analysis code (which was adapted for this work), Drs. Chi Hong Tseng and Sitaram Vangala for advice on statistical analysis using mixed-effects models, Dr. Ricardo Mostany and Grant Higerd

for their early ISI experiments in stroke, and Drs. Steven Cramer, Anubhuti Goel, Nazim Kourdougli, Anand Suresh, and Bo Wang for helpful discussions.

## Author contributions

W.A.Z., M.M., and C.P.-C. designed the experiments. W.A.Z., M.M., S.S., N.N., and I.S. performed the research and analyzed the data. W.A.Z. and C.P.-C. wrote the manuscript with input from M.M.

## Competing interests

The authors declare no competing interests.
