## [Peer Review File · Nature Communications]

Reviewers' Comments:

Reviewer #1:

Remarks to the Author:

The Portera-Cailliau lab examine whether stroke leads to neurons adopting new functional roles (or simply changing response profile), often referred to as "remapping" in the stroke literature or whether neurons simply regain responsiveness. To test this idea they induce stroke in barrel cortex and examine single cell Ca⁺⁺ responses in peri-infarct cortex (ie. D3 region which surrounds C1 barrel) to stimulation of the stroke affected whisker (C1). Using IOS and Ca⁺⁺ imaging, they found no evidence of "remapping" in D3 region (~500um from infarct border) without any stroke intervention. The absence of remapping is inferred from the fact that C1 whisker stimulation (which normally activates ~15% D3 neurons) does not drive more D3 neural responses after stroke. Instead there is a loss of responsiveness that does not fully recover, similar to that reported by previous longitudinal Ca⁺⁺ imaging studies (Tennant et al., 2017). The authors then use a cFos reporter approach and find similar proportion of C1 whisker activated cells in peri-infarct cortex, thereby confirming their previous observation. Therefore, they then try forced use therapy by plucking whiskers except C1 from the affected side of the face. This manipulation did however, lead to some reactivation of peri-infarct cortex at 2, but not 1 month of recovery based on IOS imaging. Similar to before, they did not find a greater fraction of D3 neurons driven by C1 whisker stimulation. Based on these experiments, they conclude there is no functional remapping of C1 to new circuits. Instead of remapping, they find that D3 neurons that were already driven by C1 whisker, can become more reliably activated after stroke in conjunction with forced use therapy.

Overall, I thought the paper was well written, with a nice logical flow and clear hypotheses presented and tested. The experiments and data analysis are conducted in an expert manner (and believable) and support the primary conclusions of the paper, albeit with some caveats about generalizability to other cortical regions. Below are some comments that are relatively minor in nature, mostly involving data analysis and do not require any new experiments.

Comments:

1) Regarding IOS imaging, the authors show thresholded cortical region responses after stroke in Fig 1 but this may mask lower amplitude responses. It appears that manual thresholding was used post-stroke to create C1 evoked maps but that seems like an atypical approach. Is it not possible for a standardized, quantitative approach? Further, can the authors please show/quantify delta R responses in ROI's in peri-infarct cortex at 13d and 1 month, as well as map area in Figure 1, similar to what they show in Figure 4?

2) The stimulation paradigm consisted of 1s stimulation followed by 3 sec no stim, essentially meaning that neurons were in a "stimulated" state for 25% of the time. Given that forced use therapy increases spontaneous activity, how can the authors control for the fact that by chance alone, many "responses" during each sweep may be spontaneous and not evoked in nature, thereby complicating the interpretation of Fig 6G?

3) Could the authors please show the sham stroke control data (from the first experiment) for metrics presented in Fig 5 and 6? I think it is helpful to compare to the no-stroke condition as well.

4) For Ca⁺⁺ imaging, approximately how far from the infarct border was imaging done? Previous studies have found a strong relationship with distance, including the Tennant et al., paper and previous Portera-Cailliau papers.

5) While not the focus of the study, the transient increase in D3 neuron responses to D3 whisker 5 days after C1 stroke (Fig 2), is quite interesting. It suggests there may be some ephemeral reduced or dis-inhibition of spared circuits (perhaps from stroke related disruption of cross column inhibitory connections from C1 barrel). Reminds me of work from Dan Simons showing rapid loss of lateral inhibition in barrels surrounding that corresponding to the plucked whisker (Kelly et al., JNsci, 1999). Did the authors also observe this in the forced use experiments?

6) For Fig. 3, I think it makes sense to include Supp Figure 1 data in Main Fig 3 since the layer specific differences was the first thing that came to mind, especially given my comment above. Also for readability, please indicate approximate layer boundaries in Fig. 3.

7) While I'm very supportive of this paper and I think the data clearly backs their conclusions, the

data do not prove that "remapping" could not occur in different cortical regions. Therefore, the title could be more conservative and reflect this by stating: "Barrel cortical stroke plasticity involves potentiating..." Furthermore, the term "remapping" for meso/macrosopic imaging is based not only on activity patterns of within column layer 2/3 neurons (that were imaged in the present study), but further propagation of activity, horizontally and vertically between/within cortical columns/areas over time. In this sense, the potentiation of existing layer 2/3 neurons spared by stroke described in this study could contribute to meso/macrosopic evidence of "remapped" cortical activity in peri-infarct cortex (based on VSD, GCaMP, even IOS imaging in other studies), where "remapping" is defined by the area/amplitude of cortex activated, not necessarily by what cells initiated this population change in area/amplitude. Something to consider as a discussion point since some people view the concept of remapping with more greys, than a binary, black/white phenomenon.

Reviewer #2:

Remarks to the Author:

Barrel cortex stroke and remapping, a high quality study that addresses an important question and will be of wide interest. The authors convincingly show a lack (or little) functional remapping in barrel cortex of mice subjected to photothrombotic stroke. The authors add clever fos-trap experiments and show a new take on this question, although again negative data showing little off-barrel activation. The work here goes against most studies where re-mapping of function has been described using macroscopic imaging methods.

I am familiar with several of the papers they cite and do offer some explanations for the differences. I have little doubt in the quality of the data presented here, but do differ in opinion on its interpretation. I offer some comments that would help to strengthen these conclusions.

The author should make it clear that they have not actually measured structural remapping. What they have examined is functional remapping under specific conditions and for a particular brain region. The title should have barrel cortex in it and the abstract should outline that calcium imaging was done in the D3 barrel in response to C1 whisker stimulation. I think at the least the authors need to tone down the abstract and make it more specific along with the title implicating barrel cortex, not all cortical plasticity.

While the barrel cortex is interesting from an anatomical standpoint and offers powerful manipulations, there are some potential limitations. First of all the percentage of mixed (2 barrels) activation is quite high in the baseline condition at the cellular level. I guess the effects of this get thresholded out in their intrinsic signal maps?

In other single neuron reorganization studies forelimb and hind limb responses were potentially more selective so new activations were more impactful over a background of very little cross talk. It is possible that the limbs could be different from the whiskers.

While the proportion of mixed activation is high (single neuron) the presentation of the intrinsic signal maps makes it look like there is no activation in the space between the chosen barrels in Fig. 1. The authors should show plots of the raw intrinsic data to illustrate that there is indeed overlap in functional activation between the whiskers chosen in terms of their cortical activation (supplement).

The intrinsic maps should be made at range of thresholds (supplemental), the values for the threshold clearly indicated in the legends. "For qualitative assessment of whisker-evoked ISI maps, raw ISI signals were manually thresholded to create a binary mask." If they wish to make a quantitative argument the thresholds need to be fixed or at least indicated.

Could they have chosen whisker pairs that would have had more overlapping representations to begin with, for example c2 and c1?

Can they comment about the contralesional hemisphere, they should have this data for at least the fos-trap

The author's employ anesthesia in measuring neuronal cell activity. This could be a potential confound and needs to be addressed, potentially the high percentage of salt and pepper neurons could be potentially elevated by this form of anesthesia? This could be a discussion point

The last figure showing increases in response reliability could be a reflection of some re-mapping of inputs. Although, unique neurons were not observed there were detectable increases in strength. This is potentially consistent with remapping and would be a reason to be more conservative about the interpretations.

Figure 4 panel D looks reminiscent of many remapping studies, Culver and others come to mind by intrinsic signal and GCAMP. The authors need remember that the remapped activations we're never of amplitude equal to the native response. Given almost half the animals show some map it would be important to show maps as supplemental data for all animals as in Figure 4. It would also have been interesting to examine the re-mapped weak signal area in 4D at the 2-photon level? Is there any guided 2P imaging based on these intrinsic signal maps?

In figure 5C there are many neurons that respond to C1 whisker stimulation but no map is seen by the intrinsic signal imaging this seems strange or to indicate that the method lacks sensitivity in the authors hands?

The changes in the control group in 5C in terms of % active neurons is quite high, the statistics should compare the fraction responding after stroke at 5 days to the last time points too? The fraction of responsive cells even recovers to the point where it is not significantly different.

Reviewer #3:

Remarks to the Author:

Based on years of research claiming that following injury (here photothrombotic stroke) to cortical tissue surrounding areas to the injured tissue are remapped to assume the roles of the tissue that died, the authors tested that claim in the somatosensory cortex of the mouse, by experimentally destroying the C1 barrel. The authors, using several techniques including intrinsic signal imaging, TRAP labeling, and 2-photon microscopy of neurons applied to whisker cortical representation, the authors report that they could not detect the expected evidence for remapping. In addition, the authors report that removing all whiskers by plucking except the one that corresponds to the missing barrel resulted in a different plasticity: enhanced response in neurons that were already responsive to that whisker at baseline.

While I believe that it is always healthy and even welcome to revisit consensus beliefs in any branch of science, the current manuscript suffers from some major methodological and interpretation issues that seem to undermine the authors' strong statements about their findings.

Rodents use their whiskers to scan their environment with a frequency range of 5-10 Hz. It is therefore quite unclear why the authors decided to use the unnatural 100 Hz stimulation as their routine stimulation. This puzzling choice has many ramifications that may have influenced their findings. There are not many neurons in somatosensory cortex, known for its sparse firing patterns of its neurons, that could follow or respond a 100 Hz stimulation. This could explain for example the surprising low percentage of responding neurons even in sham animals (~10% for C1 stimulation Fig. 2D or ~35% figure 2C), where one expects figures based on electrophysiological

studies is about 70-90%. It is even less clear why there is a significant difference between C1 sham stimulation (~10%) and D3 sham stimulation (~35%). In addition, the authors should have shown, using ISI, how a response to 100 Hz stimulation looks like, so there is some baseline to understand their results, especially as they use an uncommon way of analyzing their ISI data, especially the manual thresholding part and the background normalization part.

The authors keep on claiming that they destroyed the C1 barrel. In reality, the authors destroyed a major volume of cortical tissue that also includes the C1 barrel. And the large destruction volume also includes the dysgranular zone surrounding the C1 barrel and some parts of all the first order neighboring barrels (Fig. 1c). This fact, unfortunately, has major implications for the interpretation of their finding as it is clearly not the case of just destroying the C1 barrel (and in other cases the authors interpret their results as only a partial destruction of the C1 barrel, pointing to a very variable outcome of their lesions) . In addition, the authors used whisker D3 that is not a first order neighboring whisker to C1 as their test whisker, which could influence their findings because it is the first order neighboring neurons that should show the strongest remapping, not the second order neurons. Therefore, the authors' choices of stimulation frequency, the volume of lesion and location of the test whisker have, in my opinion, weakens the interpretation of their results.

The authors claim that plucking whiskers and leaving only one remaining whisker is equivalent to 'force use' cases in human stroke therapy. The authors should do a better job in explaining the reader why this plucking strategy is an equivalent situation to force use in humans where a good arm is forced to lose its mobility. If so, why not plucking all whiskers in the ipsilesional side? Comparing single whiskers to entire hands is not convincing.

The authors report that they used both male and female mice (line 389). Did the authors demonstrate that there was no difference between male and female mice before pooling their results together?

It is unclear why the authors use an enriched environment for one type of experiments and regular cages for other experiments.

Plucking whiskers could be problematic (especially 3 times a week (lines 457-458)) as the plucking process could strongly activate the cortex and influence the findings.

Minor:

Line 98: the reference (38) does not seem to be related to the topic of the sentence.

Dear Reviewers,

Thank you for the opportunity to respond to the issues raised. Below you will find our point-by-point rebuttal. We feel that with your guidance our manuscript has improved significantly and hope that you will find our efforts to respond to your comments satisfactory.

Reviewer #1 (Remarks to the Author):

The Portera-Cailliau lab examine whether stroke leads to neurons adopting new functional roles (or simply changing response profile), often referred to as “remapping” in the stroke literature or whether neurons simply regain responsiveness. To test this idea they induce stroke in barrel cortex and examine single cell Ca⁺⁺ responses in peri-infarct cortex (ie. D3 region which surrounds C1 barrel) to stimulation of the stroke affected whisker (C1). Using IOS and Ca⁺⁺ imaging, they found no evidence of “remapping” in D3 region (~500um from infarct border) without any stroke intervention. The absence of remapping is inferred from the fact that C1 whisker stimulation (which normally activates ~15% D3 neurons) does not drive more D3 neural responses after stroke. Instead there is a loss of responsiveness that does not fully recover, similar to that reported by previous longitudinal Ca⁺⁺ imaging studies (Tennant et al., 2017). The authors then use a cFos reporter approach and find similar proportion of C1 whisker activated cells in peri-infarct cortex, thereby confirming their previous observation. Therefore, they then try forced use therapy by plucking whiskers except C1 from the affected side of the face. This manipulation did however, lead to some reactivation of peri-infarct cortex at 2, but not 1 month of recovery based on IOS imaging. Similar to before, they did not find a greater fraction of D3 neurons driven by C1 whisker stimulation. Based on these experiments, they conclude there is no functional remapping of C1 to new circuits. Instead of remapping, they find that D3 neurons that were already driven by C1 whisker, can become more reliably activated after stroke in conjunction with forced use therapy. Overall, I thought the paper was well written, with a nice logical flow and clear hypotheses presented and tested. The experiments and data analysis are conducted in an expert manner (and believable) and support the primary conclusions of the paper, albeit with some caveats about generalizability to other cortical regions. Below are some comments that are relatively minor in nature, mostly involving data analysis and do not require any new experiments.

Comments:

1) Regarding IOS imaging, the authors show thresholded cortical region responses after stroke in Fig 1 but this may mask lower amplitude responses. It appears that manual thresholding was used post-stroke to create C1 evoked maps but that seems like an atypical approach. Is it not possible for a standardized, quantitative approach? Further, can the authors please show/quantify delta R responses in ROI's in peri-infarct cortex at 13d and 1 month, as well as map area in Figure 1, similar to what they show in Figure 4?

This is an important point. It's important to note that the manual thresholding was only done for the display images; all of our quantitative analyses had been done on ISI signal intensity. Based on this feedback, we have re-analyzed all of our ISI imaging data using an automated thresholding method. We calculated standardized Z-scores of raw change in reflectance values (see Methods lines 456-458) and found that a threshold of Z<-3 isolated ISI signal maps well compared to background (see new supplementary figure Fig. S2). We then applied this Z-score-based thresholding method to calculate map area for our two cohorts of mice (see new figures Fig. 1E and 4E). All representative ISI images (Figs. 1D, 4C-D, 5A) have been replaced with new images using this automated thresholding method. As requested, we have also added ISI mean signal intensity graphs in Figure 1 (Fig. 1F) and added the 13 d and 1 mo time points in Figure 4 (Fig. 4F).

2) The stimulation paradigm consisted of 1s stimulation followed by 3 sec no stim, essentially meaning that neurons were in a “stimulated” state for 25% of the time. Given that forced use therapy increases spontaneous

activity, how can the authors control for the fact that by chance alone, many “responses” during each sweep may be spontaneous and not evoked in nature, thereby complicating the interpretation of Fig 6G?

When examining sensory-evoked responses, we first classified neurons into stimulus-locked and non-stimulus-locked populations using a probabilistic bootstrapping method to correlate calcium transients with epochs of stimulation. The example traces in Fig. 2b show nicely how stimulus locked neurons show few if any spontaneous calcium transients during the inter-stimulus interval period. It's as if stimulus locked neurons engage in a pattern of activity during repetitive whisker stimulation when they only fire in response to deflections and exhibit no spontaneous activity in between. Furthermore, and this is perhaps not what Reviewer #1 is referring to, if one compares the sensory-evoked responses from stimulus-locked neurons to those from non-stimulus-locked neurons (Fig. 6D vs. Suppl. Fig. S5A) one can see that non-stimulus-locked neurons show no appreciable stimulus-evoked response. Since these non-stimulus-locked neurons (which were intermixed in the same population as our stimulus-locked neurons) are more likely to have spurious activity between the 1 sec epochs of stimulation ('spontaneous' calcium transients, if you will), and yet clearly show no stimulus-locked responses, we feel confident that the observed responses in stimulus-locked neurons are not recorded by chance and do reflect stimulus-evoked activity.

3) Could the authors please show the sham stroke control data (from the first experiment) for metrics presented in Fig 5 and 6? I think it is helpful to compare to the no-stroke condition as well.

We have included stimulus-evoked responses for D3-whisker stimulation as Reviewer Figure 1 below. We found no significant effect of group, time, or group:time interaction on D3-whisker evoked responses after stroke. However, given the variability in the data, the control group is likely somewhat underpowered for this analysis (N=6), and so we have not included this data in the manuscript.

Reviewer Figure 1. D3-whisker evoked responses are stable after C1-targeted stroke.

Quantification of the AUC from the mean D3 whisker-evoked response of neurons in the D3 barrel with D3 stimulus-locked responses after stroke targeting the C1 barrel. LME model, main effects of group (G, $p=0.130$), timepoint (T, $p=0.208$) and group-by-timepoint interaction (G:T, $p=0.702$).

4) For Ca^{++} imaging, approximately how far from the infarct border was imaging done? Previous studies have found a strong relationship with distance, including the Tennant et al., paper and previous Portera-Cailliau papers.

At baseline, imaging FOVs for surround barrels were an average distance of $473 \pm 11 \mu\text{m}$ from the center of the C1 barrel FOV to the center of the surround barrel FOV (range: $340\text{-}674 \mu\text{m}$). After stroke, we sought to image the same FOV if possible. In some cases, we could not image in exactly the original FOVs due to

damage from the stroke or stroke-induced changes in vasculature. In these cases we imaged as close as possible to the original FOV, which was typically immediately adjacent to the infarct border. We have updated the text of the manuscript with this information (see lines 518-523).

5) While not the focus of the study, the transient increase in D3 neuron responses to D3 whisker 5 days after C1 stroke (Fig 2), is quite interesting. It suggests there may be some ephemeral reduced or dis-inhibition of spared circuits (perhaps from stroke related disruption of cross column inhibitory connections from C1 barrel). Reminds me of work from Dan Simons showing rapid loss of lateral inhibition in barrels surrounding that corresponding to the plucked whisker (Kelly et al., JNsci, 1999). Did the authors also observe this in the forced use experiments?

We agree, this is an interesting observation. However, we were unable to test D3 whisker-evoked responses in the forced use experiments because all whiskers (including D3) contralateral to the stroke except the C1 whisker were plucked. We now include the Kelly et al. reference in the paper.

6) For Fig, 3, I think it makes sense to include Supp Figure 1 data in Main Fig 3 since the layer specific differences was the first thing that came to mind, especially given my comment above. Also for readability, please indicate approximate layer boundaries in Fig. 3.

Thank you for this suggestion. We have moved Suppl. Fig. 1 to main Fig. 3C as requested. We have also indicated approximate layer boundaries in Fig. 3B.

7) While I'm very supportive of this paper and I think the data clearly backs their conclusions, the data do not prove that "remapping" could not occur in different cortical regions. Therefore, the title could be more conservative and reflect this by stating: "Barrel cortical stroke plasticity involves potentiating..." Furthermore, the term "remapping" for meso/macroscopic imaging is based not only on activity patterns of within column layer 2/3 neurons (that were imaged in the present study), but further propagation of activity, horizontally and vertically between/within cortical columns/areas over time. In this sense, the potentiation of existing layer 2/3 neurons spared by stroke described in this study could contribute to meso/macroscopic evidence of "remapped" cortical activity in peri-infarct cortex (based on VSD, GCaMP, even IOS imaging in other studies), where "remapping" is defined by the area/amplitude of cortex activated, not necessarily by what cells initiated this population change in area/amplitude. Something to consider as a discussion point since some people view the concept of remapping with more greys, than a binary, black/white phenomenon.

Yes, we agree that these are important points to discuss, and our evidence is not definitive. We have changed the title as suggested. We have also added additional discussion of the limitations of our approach and the nuances of studying remapping at different scales to the discussion (see lines 343-354).

Reviewer #2 (Remarks to the Author):

Barrel cortex stroke and remapping, a high quality study that addresses an important question and will be of wide interest. The authors convincingly show a lack (or little) functional remapping in barrel cortex of mice subjected to photothrombotic stroke. The authors add clever fos-trap experiments and show a new take on this question, although again negative data showing little off-barrel activation. The work here goes against most studies where re-mapping of function has been described using macroscopic imaging methods.

Yes, we think it has become a de-facto assumption that remapping occurs after all types of strokes, even though careful studies with single-cell resolution recordings before and after stroke are lacking.

I am familiar with several of the papers they cite and do offer some explanations for the differences. I have little doubt in the quality of the data presented here, but do differ in opinion on its interpretation. I offer some

comments that would help to strengthen these conclusions.

The author should make it clear that they have not actually measured structural remapping. What they have examined is functional remapping under specific conditions and for a particular brain region. The title should have barrel cortex in it and the abstract should outline that calcium imaging was done in the D3 barrel in response to C1 whisker stimulation. I think at the least the authors need to tone down the abstract and make it more specific along with the title implicating barrel cortex, not all cortical plasticity.

We have changed the title as suggested. We agree particularly with the idea that the term 'remapping' has been applied somewhat broadly (and loosely) to post-stroke changes on multiple scales and thus it is important to specify that we measured functional remapping of individual neurons. In the Abstract we have clarified the sites of imaging and mention the caveat that our results are limited to barrel cortex stroke and related somatosensory recovery (see Abstract and lines 331-336, 408-411)

While the barrel cortex is interesting from an anatomical standpoint and offers powerful manipulations, there are some potential limitations. First of all the percentage of mixed (2 barrels) activation is quite high in the baseline condition at the cellular level. I guess the effects of this get thresholded out in their intrinsic signal maps?

Selectivity of L2/3 neurons for surround whiskers in barrel cortex tends to be significantly higher than in other areas of somatosensory cortex or even in L4 of barrel cortex. The percentage of neurons responding to surround whiskers (for example C1 responsive neurons in the D3 barrel) in L2/3 we observed is largely similar to other two-photon imaging studies using passive whisker deflection in barrel cortex (see Clancy et al. *J. Neurosci.* 2015). It is likely that we do not see a C1-whisker evoked ISI signal in the D3 barrel (for example) because, although some C1-whisker responsive neurons are present in L2/3, these are not numerous enough to generate the hemodynamic changes that form the basis of ISI signals in the absence of strong excitatory inputs into L4 (which are more tightly linked to the anatomical barrel column).

In other single neuron reorganization studies forelimb and hind limb responses were potentially more selective so new activations were more impactful over a background of very little cross talk. It is possible that the limbs could be different from the whiskers.

We agree that this is an important consideration and valid caveat to our findings. We have added this to the discussion (see lines 331-336) and now include the word "barrel" in the title.

While the proportion of mixed activation is high (single neuron) the presentation of the intrinsic signal maps makes it look like there is no activation in the space between the chosen barrels in Fig. 1. The authors should show plots of the raw intrinsic data to illustrate that there is indeed overlap in functional activation between the whiskers chosen in terms of their cortical activation (supplement).

We have added raw ISI data for the C1 and D3 barrels from two representative mice as new Suppl. Fig. 2, which nicely illustrates that the activity maps elicited by C1 and D3 whisker stimulation do not overlap, even examining a range of threshold values (see Suppl. Fig. 2B and D). In fact, this is why we chose these particular whiskers – so that we could ablate the entire C1 activity map (which does extend slightly beyond the anatomical C1 barrel; see new Fig. S1), while sparing the D3 activity map. As indicated above, the lack of a C1 whisker-evoked ISI signal in the D3 barrel despite the presence of some L2/3 neurons responsive to the C1 whisker is likely due to the fact that ISI signals depend on strong excitatory inputs to L4 generating a significant hemodynamic response throughout the cortical column, rather than on the sparsely active whisker-responsive neurons present in L2/3. We have no doubt that stimulating neighboring whiskers would elicit overlapping maps.

The intrinsic maps should be made at range of thresholds (supplemental), the values for the threshold clearly indicated in the legends. "For qualitative assessment of whisker-evoked ISI maps, raw ISI signals were

manually thresholded to create a binary mask." If they wish to make a quantitative argument the thresholds need to be fixed or at least indicated.

As noted in response to Reviewer #1's first comment above, we have re-analyzed all of our ISI imaging data using an automated thresholding method, using the same standard threshold of $Z < -3$ for the quantification of map area throughout the manuscript.

Could they have chosen whisker pairs that would have had more overlapping representations to begin with, for example c2 and c1?

Yes, when we first embarked on this study, we considered several potential whisker pairs. We suspected that remapping might be most obvious in the barrel immediately adjacent to C1. However, we found that trying to achieve a small enough infarct that spared surrounding barrels was technically not feasible (small infarcts did not consistently obliterate the C1 map). In fact, our stroke size is already on the small end of the range of stroke sizes observed in humans. Had we chosen two adjacent barrels, the stroke would have affected them both. Thus, we felt that smaller sized strokes might be less translationally relevant. Given the salt & pepper distribution of single neuron responses to their surrounding barrels, we were confident that if remapping were to occur after stroke, we would still be able to see it in the D3 barrel. Therefore, in the end, we chose the C1/D3 pair to allow us to produce consistent infarcts, to be within the range of typical human stroke sizes, and yet still sparing significant portions of barrel cortex to visualize plasticity if indeed it occurred.

Can they comment about the contralesional hemisphere, they should have this data for at least the fos-trap

Unfortunately, we did not examine TRAP data for the contralesional hemisphere. The rationale for this was that in our previous studies we found no evidence of plasticity (structural or functional) in the contralesional hemisphere following a much larger stroke that destroyed the entire S1BF (see Johnston et. al *Cereb. Cortex* 2013). We feel it is unlikely that the small single barrel strokes would trigger contralateral remapping.

The author's employ anesthesia in measuring neuronal cell activity. This could be a potential confound and needs to be addressed, potentially the high percentage of salt and pepper neurons could be potentially elevated by this form of anesthesia? This could be a discussion point

Yes, we agree that anesthesia is a potential caveat that we now mention in the discussion (see lines 418-421). However, sparse coding of whisker inputs in S1BF (salt & pepper distribution of neurons in L2/3) is a feature of both awake and anesthetized mice (see for example O'Connor et. al *Neuron* 2010; Crochet et. al *Neuron* 2011; and Peron et. al 2015, among others). We have also recently compared whisker evoked activity in awake and anesthetized mice and find no difference in the percentage of stimulus-locked neurons in a given barrel (not shown here). Keep in mind that the level of anesthesia was quite low (0.5% isoflurane) because mice were also sedated with chlorprothixene. For this paper we chose to record under this light anesthesia because some unanesthetized mice tend will grab the piezo stimulator, which obviously affects the recording. Finally, our activity-dependent labeling experiments using TRAP mice were done in fully awake, behaving animals without any anesthesia. These experiments confirmed our main finding that C1 whisker-responsive neurons do not increase in number in the peri-infarct cortex after stroke.

The last figure showing increases in response reliability could be a reflection of some re-mapping of inputs. Although, unique neurons were not observed there were detectable increases in strength. This is potentially consistent with remapping and would be a reason to be more conservative about the interpretations.

We agree that the observed increase in sensory-evoked responses is likely a result of synaptic plasticity and structural remodeling in the peri-infarct cortex as a result of the stroke. However, we do not consider this 'remapping'. Perhaps it is semantics, but we consider remapping to be a clear change in tuning, such that a given neuron will change its functionality after stroke (it acquires the ability to respond to multiple whiskers: its original whisker and the whisker corresponding to the infarcted barrel). It is an important distinction, of course, as many investigators use the term remapping more generally to describe any form of post-stroke plasticity, including changes in structure or activity after stroke. We have tried emphasize in the Abstract and Discussion

that we were measuring functional 'remapping' of individual neurons. If a neuron was already responding to the C1 whisker before stroke, and then continues to do so after the stroke only more strongly, we do not consider that to represent remapping. We have added discussion points about potential structural and macroscopic circuit remapping to the discussion (see lines 318-322 and 343-354).

Figure 4 panel D looks reminiscent of many remapping studies, Culver and others come to mind by intrinsic signal and GCAMP. The authors need remember that the remapped activations we're never of amplitude equal to the native response. Given almost half the animals show some map it would be important to show maps as supplemental data for all animals as in Figure 4.

We too thought, initially, that the small ISI maps in Fig, 4 might represent remapping. But we now strongly believe that those represent residual maps due to incomplete strokes that failed to destroy the entire activity map. The reason the maps disappear transiently at early time points (days 5 and 13) may be due to edema from the stroke and/or the reduced sensory-evoked responses in spared C1 whisker-responsive neurons that we identified (see Fig. 6D-G). We have included raw ISI data for baseline and final imaging time points for both cohorts of mice in Reviewer Figures 2 and 3.

Reviewer Figure 2. Raw ISI maps for data presented in Figure 1.

- A.** Raw C1 whisker-evoked ISI signals at baseline (top) and 1 month (bottom) after sham stroke in control animals.
B. Raw D3 whisker-evoked ISI signals at baseline (top) and 1 month (bottom) after sham stroke in control animals. Intensity scale indicates $\Delta R/R$ reflectance values (same for panels A-D).
C. Raw C1 whisker-evoked ISI signals at baseline (top) and 1 month (bottom) after C1-targeted stroke in stroke animals.
D. Raw D3 whisker-evoked ISI signals at baseline (top) and 1 month (bottom) after C1-targeted stroke in stroke animals.

Reviewer Figure 3. Raw ISI maps for data presented in Figure 4.

A. Raw C1 whisker-evoked ISI signals at baseline (top) and 2 months (bottom) after C1-targeted stroke in control animals. Intensity scale indicates $\Delta R/R$ reflectance values.

B. Raw C1 whisker-evoked ISI signals at baseline (top) and 2 months (bottom) after C1-targeted stroke in forced use (whisker plucked) animals. Intensity scale indicates $\Delta R/R$ reflectance values, except for two animals (c328mB and c328mC) at 2 months post-stroke; for these two animals, the raw $\Delta R/R$ reflectance images (obtained > 2 years ago) could not be recovered, so intensity scaled (as a percentage of maximum) images were used instead.

It would also have been interesting to examine the re-mapped weak signal area in 4D at the 2-photon level? Is there any guided 2P imaging based on these intrinsic signal maps?

For animals with an apparent ISI signal, we did try to perform two-photon imaging in these areas. However, when comparing the percentage of C1 whisker-responsive neurons at 2 months between FOV with or without ISI signal, we found no significant effect of group, FOV, or group:FOV interaction (see Reviewer Fig. 4). This may reflect different sources of signals for ISI and 2P imaging and/or impairments in neurovascular coupling post-stroke (since ISI signals are largely dependent on the hemodynamic response).

Reviewer Figure 4. Effect of field-of-view on percentage of stimulus- locked cells.

Percentage of peri-infarct neurons with stimulus-locked responses to C1-whisker stimulation at 2 months post-stroke in control and forced use (whisker plucked) groups. Imaging fields of view (FOV) with C1-whisker evoked ISI signals (“Map Present”) were compared to FOV in areas of S1BF without detectable C1-whisker evoked ISI signal (“Map Absent”). Two-way ANOVA, main effects of group (G, $p= 0.713$), FOV (F, $p= 0.375$), and group-by-FOV interaction (G:F, $p<0.253$) were not significant.

In figure 5C there are many neurons that respond to C1 whisker stimulation but no map is seen by the intrinsic signal imaging this seems strange or to indicate that the method lacks sensitivity in the authors hands?

As described above, we believe the relatively low percentage of C1 whisker-responsive L2/3 neurons in surround barrels is insufficient to generate an ISI map; note that ISI maps are generated mainly by hemodynamic responses driven by strong excitatory inputs into L4 neurons and, after stroke, there are not enough strong excitatory inputs for the C1 barrel to generate the hemodynamic response required for an ISI signal. In addition, to perform ISI imaging, we focus down ~300 μm from the cortical surface (i.e. L4) to avoid surface blood vessel artifacts. We certainly do not believe the ISI technique lacks sensitivity in our hands, as we were able to generate consistent single whisker ISI maps across months. In fact, this is a somewhat unique aspect of our study, that we could reliably detect single whisker maps over periods of weeks before and after stroke. In addition, we have previously used ISI to monitor rather subtle changes in single whisker map size as a result of whisker plucking or between mouse genotypes (see He et. al *Mol. Psychiatry* 2019 and Gao et. al *Elife* 2017). Thus, we are confident in the sensitivity of ISI in our hands.

The changes in the control group in 5C in terms of % active neurons is quite high, the statistics should compare the fraction responding after stroke at 5 days to the last time points too? The fraction of responsive cells even recovers to the point where it is not significantly different.

This is correct – in Fig. 5C, the proportion of C1 whisker-responsive neurons recovers to baseline levels at 1 and 2 mo post-stroke and is not statistically significantly different from baseline. This result stands in contrast to the classical remapping hypothesis which would predict that the proportion of C1-whisker responsive neurons would increase well above baseline levels as a result of functional remapping and recruitment of spared neurons to become newly C1 whisker responsive. This result was seen both in control animals (with C1

barrel targeted stroke and no intervention) and whisker plucked animals (C1 barrel targeted stroke plus whisker plucking). For statistical analysis we used a generalized linear mixed model approach (guided by statistical consultants at UCLA), which overcomes some of the limitations of repeated measures ANOVAs (assumption of normality, need for balanced groups, intolerance of missing data), but does not yield all possible statistical pairwise comparisons as one might compute for a repeated measures ANOVA. Re-analyzing the data using +5 d as the intercept instead of baseline, we find a p-value of 0.01 for the 2 mo timepoint, suggesting the proportion of stimulus locked neurons has significantly increased from post-stroke day 5 to 2 months. Regardless, this comparison does not change the main conclusion of the experiment, which is that the proportion of stimulus locked neurons decreases acutely following stroke before recovering to baseline levels (but not above it) at 1-2 mo post stroke. Hopefully this answer satisfies the Reviewer's query.

Reviewer #3 (Remarks to the Author):

Based on years of research claiming that following injury (here photothrombotic stroke) to cortical tissue surrounding areas to the injured tissue are remapped to assume the roles of the tissue that died, the authors tested that claim in the somatosensory cortex of the mouse, by experimentally destroying the C1 barrel. The authors, using several techniques including intrinsic signal imaging, TRAP labeling, and 2-photon microscopy of neurons applied to whisker cortical representation, the authors report that they could not detect the expected evidence for remapping. In addition, the authors report that removing all whiskers by plucking except the one that corresponds to the missing barrel resulted in a different plasticity: enhanced response in neurons that were already responsive to that whisker at baseline.

While I believe that it is always healthy and even welcome to revisit consensus beliefs in any branch of science, the current manuscript suffers from some major methodological and interpretation issues that seem to undermine the authors' strong statements about their findings.

Rodents use their whiskers to scan their environment with a frequency range of 5-10 Hz. It is therefore quite unclear why the authors decided to use the unnatural 100 Hz stimulation as their routine stimulation. This puzzling choice has many ramifications that may have influenced their findings. There are not many neurons in somatosensory cortex, known for its sparse firing patterns of its neurons, that could follow or respond a 100 Hz stimulation. This could explain for example the surprising low percentage of responding neurons even in sham animals (~10% for C1 stimulation Fig. 2D or ~35% figure 2C), where one expects figures based on electrophysiological studies is about 70-90%. It is even less clear why there is a significant difference between C1 sham stimulation (~10%) and D3 sham stimulation (~35%). In addition, the authors should have shown, using ISI, how a response to 100 Hz stimulation looks like, so there is some baseline to understand their results, especially as they use an uncommon way of analyzing their ISI data, especially the manual thresholding part and the background normalization part.

We can address all of these concerns by clarifying a few things that we perhaps failed to make clearer, and we apologize for this. For intrinsic signal imaging, we chose 100 Hz stimulation simply because, in our experience, it yields more robust ISI maps than lower frequencies closer to the natural whisking range. There is definitely evidence that S1BF neurons can encode vibration frequencies up to several hundred Hz (see Ewert et. al J Neurosci 2008). Importantly, after additional experiments we found no difference in map location comparing 10 Hz to 100 Hz stimulation and we have included this data as Reviewer Fig. 5 below. Importantly, we want this Reviewer to know that we used 10 Hz stimulation for all two-photon imaging experiments.

Reviewer Figure 5. 100 Hz whisker stimulation produces more robust ISI maps.

Representative data from two mice showing C1 whisker-evoked ISI maps obtained with 10 Hz (left panels) or 100 Hz (middle panels) whisker stimulation. Binary images were generated from raw ISI signals using a Z-score threshold of -2.5. Right panels: 10 Hz (green) and 100 Hz (magenta) activity maps are overlaid on photographs of the cranial window to demonstrate that both frequencies of stimulation produce maps at the same location. Scale bar = 0.5 mm.

Regarding the comment about manual thresholding, we want to emphasize that we never analyzed data based manual thresholds (those has only been used for display images); we had analyzed map intensity differences from raw images. In response to a similar misunderstanding by the other Reviewers, we have now re-analyzed our ISI data using an automated thresholding method (now shown in Figs. 1D, 4C-D, 5A).

Regarding the proportion of stimulus locked neurons, our results are certainly consistent with several other studies using two-photon calcium imaging. For example, in assessing responsivity of L2/3 neurons in S1BF induced by passive deflection of single whiskers Dan Feldman's lab found that 25% of neurons were tuned to the principal whisker (Clancy et al. *J. Neurosci.* 2015). Similar sparse coding in L2/3 has been shown for active whisking tasks (see for example O'Connor et. al *Neuron* 2010; Crochet et. al *Neuron* 2011; and Peron et. al *Neuron* 2015, among others). It is certainly conceivable that our calcium imaging approach, with its suboptimal single spike detection, might be underestimating the percentage of whisker responsive L2/3 neurons. It is also the proportion of stimulus locked neurons is significantly higher in L4 compared to L2/3. But we are not aware of any electrophysiological studies that could have recorded from dozens of L2/3 neurons simultaneously to yield the values that the Reviewer states. Keep in mind that in Fig. 2C-D, in both panels we are imaging the same population of neurons within the D3 barrel and stimulating either the D3 whisker or the C1 whisker. Thus, the lower proportion of C1 whisker-responsive neurons compared to D3 whisker-responsive neurons is expected given that D3 is the principal whisker for the barrel, whereas C1 is not. This distance-dependent decrease in the proportion of whisker-responsive neurons for a given whisker (as one moves away from the barrel corresponding to the principal whisker) is well established (see again Clancy et al.). When we quantify C1 whisker-responsive neurons in the C1 barrel (see Fig. 5B) we find a very similar proportion of whisker-responsive neurons as seen in the D3 barrel with D3 whisker stimulation (Fig. 2C).

The authors keep on claiming that they destroyed the C1 barrel. In reality, the authors destroyed a major volume of cortical tissue that also includes the C1 barrel. And the large destruction volume also includes the dysgranular zone surrounding the C1 barrel and some parts of all the first order neighboring barrels (Fig. 1c).

This fact, unfortunately, has major implications for the interpretation of their finding as it is clearly not the case of just destroying the C1 barrel (and in other cases the authors interpret their results as only a partial destruction of the C1 barrel, pointing to a very variable outcome of their lesions).

This is actually not accurate that our stroke involved a major volume of cortical tissue, but in retrospect we should have done a better job at showing the extent of damage to the S1BF, as this is an important point. In the manuscript, we had tried to indicate that strokes were targeted to the C1 barrel with the goal of ablating the C1 whisker activity map, which certainly includes some damage to the immediately adjacent barrels as well. This was depicted graphically in Fig. 1A and 4A. We have now made this point even more explicit in the text. Importantly, we have also added a new Suppl. Fig. 1 (also shown here as Reviewer Fig. 6) that depicts tangential sections through L4 after C1 barrel-targeted stroke. This figure clearly shows that the C1 barrel is ablated along with some damage to immediately adjacent barrels. Critically, this figure also shows that the collateral damage does not extend into the dysgranular zone posterior to the S1BF. In addition, we have added some text to the discussion noting the caveat that damage to the dysgranular zone could potentially affect remapping (see lines 358-364).

Reviewer Figure 6. Targeting of photothrombotic strokes to the C1 barrel. (same as Supp. Fig. 1)

A. Tangential section through layer 4 of the S1BF 5 days after photothrombotic stroke targeting the C1 barrel. Scnn1a-Tg3-Cre mice crossed with Ai162(TIT2L-GC6s-ICL-tTA2)-D mice were used to express GCaMP6s in layer 4 and highlight individual barrels in the S1BF.

B. Higher magnification of infarct area depicted in A.

C. Deeper section of layer 4 from the same mouse more clearly showing the infarct core. Scale bar = 250 μ m.

D. Higher magnification of infarct area in depicted in C. Note that the infarct does not extend beyond the barrels immediately surrounding the C1 barrel. Scale bar = 250 μ m.

In addition, the authors used whisker D3 that is not a first order neighboring whisker to C1 as their test whisker, which could influence their findings because it is the first order neighboring neurons that should show the strongest remapping, not the second order neurons. Therefore, the authors' choices of stimulation frequency, the volume of lesion and location of the test whisker have, in my opinion, weakens the interpretation of their results.

We have addressed this concern in response to Reviewer #2. Briefly, we chose the C1/D3 pair to allow us to produce infarcts of consistent size, large enough to destroy a single barrel yet sparing significant portions of surrounding barrels that the classical remapping hypothesis would propose are involved in plasticity. In the

second experiment examining the effects of whisker plucking, we imaged multiple FOV around the C1 barrel. These were typically immediately adjacent to the infarct border. The fact that we see plasticity in this peri-infarct region (on average ~470 μm away from the infarct) suggests we recorded from the appropriate regions for compensatory plasticity (it's just that there is no remapping). While it is possible that more robust remapping could have been found if more precise lesions of a single barrel had been possible (completely sparing the immediately adjacent barrels), we feel that such a scenario would not be consistent with clinical strokes, which are typically larger in volume, and respect vascular, rather than anatomical boundaries (and thus are unlikely to ever spare cortical territories so cleanly). Therefore, we favored slightly larger strokes as we felt the translational relevance to human stroke was greater.

The authors claim that plucking whiskers and leaving only one remaining whisker is equivalent to 'force use' cases in human stroke therapy. The authors should do a better job in explaining the reader why this plucking strategy is an equivalent situation to force use in humans where a good arm is forced to lose its mobility. If so, why not plucking all whiskers in the ipsilesional side? Comparing single whiskers to entire hands is not convincing.

Mice use their whiskers to explore their environment. By plucking all whiskers except the one corresponding to the infarcted barrel we wanted to recreate a scenario where human stroke patients have their 'good' arm constrained so they are forced to use their 'bad' arm. The Reviewer is correct in stating that this differs from typical forced-use therapy for hemiparesis in human stroke rehabilitation. We acknowledge it is not a perfect analogy. We had favored the term 'force use therapy' over 'constraint induced therapy' for this reason, but we agree that neither is ideal. We have added additional text to the Discussion (see lines 401-403) noting this limitation. Another advantage of using whisker plucking is that many prior studies have shown that ipsilateral whisker plucking promotes plasticity and remapping in the intact S1BF. We imagined this would enhance the chances that we observed remapping after stroke, but it wasn't so.

The authors report that they used both male and female mice (line 389). Did the authors demonstrate that there was no difference between male and female mice before pooling their results together?

While underpowered for an analysis comparing the effect of sex, we saw a trend for fewer whisker-responsive neurons in female animals at baseline, but no significant effect of sex over time. We have added this analysis as Reviewer Fig. 7 below.

Reviewer Figure 7. Effect of sex on percentage of stimulus- locked cells.

Percentage of neurons with stimulus-locked responses to C1-whisker stimulation in the peri-infarct regions throughout recovery in control and forced use whisker plucked groups, sub-grouped by sex (control male, n= 3; all other groups n= 5). GLME binomial model, main effect of sex (S, $p= 0.080$) was not significant, whereas timepoint remained significant (T, $p<0.001$). Significance for individual coefficients for timepoints (T) indicated over corresponding data points (##, $p<0.01$; ###, $p<0.001$).

It is unclear why the authors use an enriched environment for one type of experiments and regular cages for other experiments.

We apologize for the confusion. Enriched environment was used only in experiments for activity dependent labeling of C1 whisker-responsive neurons in TRAP mice, and only for 6 hours (see Fig. 3 and Methods lines 561-564). All mice in all experiments were housed in standard cages.

Plucking whiskers could be problematic (especially 3 times a week (lines 457-458)) as the plucking process could strongly activate the cortex and influence the findings.

Whisker plucking was done under deep isoflurane anesthesia in which cortical activity is likely to be significantly suppressed, so we feel it is unlikely that this acute process would strongly activate the cortex. In addition, control animals were subjected to equivalent anesthesia and manipulation (including gentle tugging) of the whiskers. We are not otherwise aware of how the whisker plucking process (a few seconds 3x/week) would influence our findings about plasticity with calcium imaging performed many hours after plucking and over a course of weeks.

Minor:

Line 98: the reference (38) does not seem to be related to the topic of the sentence.

This references our previous work using ISI to obtain single whisker evoked activity maps (the title of that paper does not make that clear though).

Reviewers' Comments:

Reviewer #1:

Remarks to the Author:

The authors have done an excellent job of thoroughly addressing the points that I raised. The additional automated analysis of IOS maps and clarification in the text regarding what "remapping" means (and limitations) have strengthened the manuscript. As a side note, I do find the lack of spontaneous activity in stimulus-locked cells very curious... maybe future studies can probe this further.

In conclusion, I have no further concerns and congratulate the authors on this impressive study.

Reviewer #2:

Remarks to the Author:

I think the manuscript is ready for publication after adding a few qualifiers to the abstract in all caps below. The authors have done a good job of responding to my comments and the work will be an important addition to this literature. I appreciate that they are open to the possibility that the barrel cortex and the whisker pairs they select are not necessarily directly comparable to the forelimb sensory and motor areas.

Using in vivo calcium imaging to longitudinally record sensory-evoked activity (UNDER ANESTHESIA), we did not find any increase in the number of C1 whisker-responsive neurons in the adjacent, spared D3 barrel after stroke.

However, definitive LONGITUDINAL evidence of neurons changing their response selectivity after stroke is lacking.

This led to an increase in the reliability of sensory-evoked responses in C1 whisker-responsive neurons but did not increase the number of C1 whisker-responsive neurons in spared surround barrels OVER BASELINE LEVELS.

Reviewer #3:

Remarks to the Author:

The authors have addressed my concerns by adding relevant experiments and clarifying issues with their original text. In my opinion, the manuscript is now ready to be published.

Dear Reviewers,

Thank you for your careful review of the revisions to our manuscript. We have made all requested changes.

Reviewer #1 (Remarks to the Author):

The authors have done an excellent job of thoroughly addressing the points that I raised. The additional automated analysis of IOS maps and clarification in the text regarding what "remapping" means (and limitations) have strengthened the manuscript. As a side note, I do find the lack of spontaneous activity in stimulus-locked cells very curious... maybe future studies can probe this further.

In conclusion, I have no further concerns and congratulate the authors on this impressive study.

We appreciate the reviewer's efforts and comments.

Reviewer #2 (Remarks to the Author):

I think the manuscript is ready for publication after adding a few qualifiers to the abstract in all caps below. The authors have done a good job of responding to my comments and the work will be an important addition to this literature. I appreciate that they are open to the possibility that the barrel cortex and the whisker pairs they select are not necessarily directly comparable to the forelimb sensory and motor areas.

Using in vivo calcium imaging to longitudinally record sensory-evoked activity (UNDER ANESTHESIA), we did not find any increase in the number of C1 whisker-responsive neurons in the adjacent, spared D3 barrel after stroke.

However, definitive LONGITUDINAL evidence of neurons changing their response selectivity after stroke is lacking.

This led to an increase in the reliability of sensory-evoked responses in C1 whisker-responsive neurons but did not increase the number of C1 whisker-responsive neurons in spared surround barrels OVER BASELINE LEVELS.

We have made the suggested edits to the abstract.

Reviewer #3 (Remarks to the Author):

The authors have addressed my concerns by adding relevant experiments and clarifying issues with their original text. In my opinion, the manuscript is now ready to be published.

Ron Frostig

We appreciate the reviewer's efforts and comments.